# Orthogonal Non-negative Tensor Factorization based Multi-view Clustering

**Jing Li**
Xidian University
Xi'an, Shaanxi, China
jinglxd@stu.xidian.edu.cn

**Quanxue Gao** *
Xidian University
Xi'an, Shaanxi, China
qxgao@xidian.edu.cn

**Qianqian Wang**
Xidian University
Xi'an, Shaanxi, China
qqwang@xidian.edu.cn

**Ming Yang**
Harbin Engineering University
Harbin, Heilongjiang, China
yangmingmath@gmail.com

**Wei Xia**
Xidian University
Xi'an, Shaanxi, China
xdweixia@gmail.com

## Abstract

Multi-view clustering (MVC) based on non-negative matrix factorization (NMF) and its variants have attracted much attention due to their advantages in clustering interpretability. However, existing NMF-based multi-view clustering methods perform NMF on each view respectively and ignore the impact of between-view. Thus, they can't well exploit the within-view spatial structure and between-view complementary information. To resolve this issue, we present orthogonal non-negative tensor factorization (Orth-NTF) and develop a novel multi-view clustering based on Orth-NTF with one-side orthogonal constraint. Our model directly performs Orth-NTF on the 3rd-order tensor which is composed of anchor graphs of views. Thus, our model directly considers the between-view relationship. Moreover, we use the tensor Schatten $p$-norm regularization as a rank approximation of the 3rd-order tensor which characterizes the cluster structure of multi-view data and exploits the between-view complementary information. In addition, we provide an optimization algorithm for the proposed method and prove mathematically that the algorithm always converges to the stationary KKT point. Extensive experiments on various benchmark datasets indicate that our proposed method is able to achieve satisfactory clustering performance.

## 1 Introduction

As one of the most typical methods in unsupervised learning, clustering has a wide scope of application [26; 4; 1] to assign data to different clusters according to the information describing the objects and their relationships. Non-negative matrix factorization (NMF) [19] is one of the representative methods of clustering, which is proved to be equivalent to K-means clustering [7]. Despite the widespread use of NMF, there are some drawbacks that have prompted some variants of NMF [8; 5; 27; 9; 3].

In particular, the one-side G-orthogonal NMF [8] can guarantee the uniqueness of the solution of matrix factorization and has excellent clustering interpretation. Also, Ding *et al.*proposed the semi-NMF [9]. The data matrix and one of the factor matrices are unconstrained, which allows semi-NMF to be more suitable for applications where the input data is mixed with positive and negative numbers.

---

*Corresponding author.

37th Conference on Neural Information Processing Systems (NeurIPS 2023).

Although the above methods can achieve outstanding clustering performance, they are all single-view clustering methods and cannot be adopted straightforwardly for multi-view clustering.

Multi-view clustering tends to achieve superior performance compared to traditional single-view clustering owing to the capability to leverage the complementary information embedded in the different views. Considering the superiority of MVC and NMF, lots of NMF-based multi-view clustering methods have been proposed [13; 28; 23; 33; 15; 29; 14; 37]. The NMF-based multi-view clustering methods can save time and space because it is unnecessary to construct affinity graphs while graph-based methods have to. However, usually, they decompose the original data matrix directly, which leads to a dramatic reduction in the efficiency of the algorithm when the dimension of the original data is huge.

Inspired by the idea of anchor graph. the above issues can be solve by carrying out NMF on the anchor graph [37]. Due to the fact that the dimension of the anchor graph is considerably smaller than the original affinity graph, it follows that the clustering efficiency can be improved. However, as is well-known, there exist two ways of NMF-based multi-view clustering methods. One is to integrate different views first and then implement the NMF on the integrated matrix; the other is to perform the NMF on different views separately and then integrate the result from each view. Both ways are essentially applications of NMF on a single view, and both need to reduce the multi-view data into matrices in the process, which causes the loss of the original spatial information.

To fix the aforesaid issues, we proposed a novel multi-view clustering based on Orth-NTF with one-side orthogonal constraint. Specifically, Non-negative Matrix Factorization (NMF) is tailored primarily for second-order matrices. When processing third-order tensors, there's a need to first transform the tensor into a matrix before applying NMF. This step can lead to a loss of inherent spatial structural information from the third-order tensor. In contrast, Non-negative Tensor Factorization (NTF) sidesteps this issue. NTF directly decomposes third-order tensors. This ensures that the NTF not only acknowledges the relationships between the views but also harnesses the complementary information they offer. Fig 1 delineates the distinction between traditional NMF-based clustering techniques and our NTF-based approach. Furthermore, by incorporating an orthogonal constraint, our model offers distinct physical interpretability for clustering. This suggests that each row of the indicator matrix contains a single non-zero element, and the position of this element directly corresponds to the label of the respective sample. A large number of experiments have shown that our methods have excellent clustering performance.

The main contributions are summarized below:

- We introduce orthogonal non-negative tensor factorization, which considers the between-view relationship directly. Also, we use tensor Schatten $p$-norm regularization to characterize the cluster structure of multi-view data and can exploit the complementary information of between views.

- We regard the anchor graph obtained from the original data as the input of the non-negative matrix factorization, which reduces the complexity of our proposed algorithm considerably.

- We provide an optimization algorithm for the proposed method and prove it always converges to the KKT stationary point mathematically. The effectiveness of its application on tensorial G-orthogonal non-negative matrix factorization is demonstrated by extensive experiments.

## 2 Related work

In recent years, multi-view clustering (MVC) has received increasing attention due to its excellent clustering performance. Also, non-negative matrix factorization (NMF) is an efficient technique in single-view clustering, which can generate excellent clustering results that are easy to interpret, and many NMF-based variants have been proposed. Therefore, multi-view clustering-based NMF and its variants have attracted tremendous interest recently.

As the first investigation of the multi-view clustering method based on joint NMF, multiNMF [23] implements NMF at each view and pushes the different clustering results of each view to a consensus. It provides a new viewpoint for the subsequent NMF-based MVC methods. Influenced by multiNMF, He *et al.*proposed a multi-view clustering method combining NMF with similarity [15]. It implements NMF on each view as in multiNMF. In addition, it sets a weight for a different view and introduces

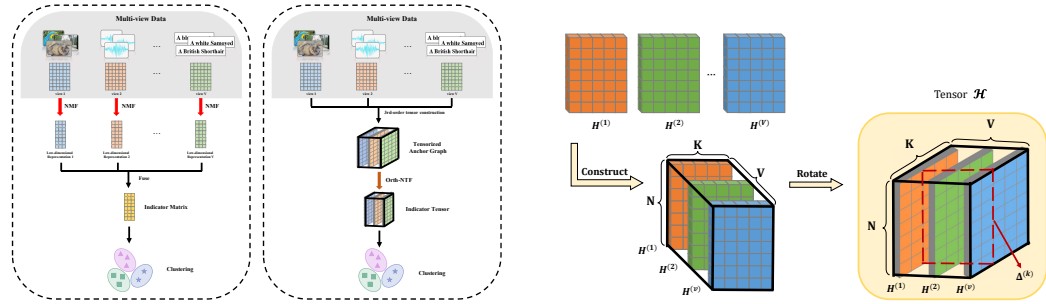

Figure 1: Comparison between traditional NMF-based multi-view clustering method (left) and Orth-NTF (right).

Figure 2: 3rd-order tensor construction process.

a similarity matrix of data points to extract consistent information from different views. To better detect the geometric structure of inner-view space, Wang *et al.* [33] introduced graph regularization into the NMF-based multi-view clustering method to improve clustering performance. Considering the above work, Wang *et al.* [29] proposed a graph regularization multi-view clustering method based on concept factorization (CF). CF is a variant of NMF and it is suitable for handling data containing negative.

As the size of data grows, lots of methods to accelerate matrix factorization are presented. Wang *et al.* [28] proposed a fast non-negative matrix triple factorization method. It constrains the factor matrix of NMF to a clustering indicator matrix, thereby avoiding the post-processing of the factor matrix. Inspired by the work of Wang, Han *et al.* [14] constrained the intermediate factor matrix in the triple factorization to a diagonal matrix, reducing the number of matrix multiplications in the solution process. Another idea to deal with large-scale multi-view data is to introduce anchor graphs, since the number of anchor points is much smaller than the number of original data, multi-view clustering methods based on anchor graphs tend to reduce the computational complexity and thus are able to deal with large-scale data [22; 17; 21]. Considering that previous NMF-based multi-view clustering methods are performed directly on the original data, Yang *et al.* [37] introduced an anchor graph as the input of G-orthogonal NMF. The efficiency of matrix factorization is indeed improved due to the introduction of anchor graph.

Despite the fact that existing NMF-based multi-view clustering methods can perform the clustering tasks excellently, they apply NMF to each view independently. Subsequently, they combine the low-dimensional representations from different perspectives to arrive at a unified shared representation. This approach often overlooks the interrelationships between the views, which are crucial for clustering.

## 3 Notations

We introduce the notations used throughout this paper. We use bold calligraphy letters for 3rd-order tensors, $\mathcal{H} \in \mathbb{R}^{n_1 \times n_2 \times n_3}$, bold upper case letters for matrices, $\mathbf{H}$, bold lower case letters for vectors, $\mathbf{h}$, and lower case letters such as $h_{ijk}$ for the entries of $\mathcal{H}$. Moreover, the $i$-th frontal slice of $\mathcal{H}$ is $\mathcal{H}^{(i)}$. $\overline{\mathcal{H}}$ is the discrete Fourier transform (DFT) of $\mathcal{H}$ along the third dimension, $\overline{\mathcal{H}} = \mathrm{fft}(\mathcal{H}, [\,], 3)$. Thus, $\mathcal{H} = \mathrm{ifft}(\overline{\mathcal{H}}, [\,], 3)$. The trace and transpose of matrix $\mathbf{H}$ are expressed as $\mathrm{tr}(\mathbf{H})$ and $\mathbf{H}^{\mathrm{T}}$. The F-norm of $\mathcal{H}$ is denoted by $\|\mathcal{H}\|_F$.

**Definition 1** (t-product [18]). *Suppose $\mathcal{A} \in \mathbb{R}^{n_1 \times m \times n_3}$ and $\mathcal{B} \in \mathbb{R}^{m \times n_2 \times n_3}$, the t-product $\mathcal{A} * \mathcal{B} \in \mathbb{R}^{n_1 \times n_2 \times n_3}$ is given by*

$$\mathcal{A} * \mathcal{B} = \mathrm{ifft}(\mathrm{bdiag}(\overline{\mathbf{A}}\overline{\mathbf{B}}), [\,], 3),$$

*where $\overline{\mathbf{A}} = \mathrm{bdiag}(\overline{\mathcal{A}})$ and it denotes the block diagonal matrix. The blocks of $\overline{\mathbf{A}}$ are frontal slices of $\overline{\mathcal{A}}$.*

**Definition 2.** *[12] Given $\boldsymbol{\mathcal{H}} \in \mathbb{R}^{n_1 \times n_2 \times n_3}$, $h = \min(n_1, n_2)$, the tensor Schatten p-norm of $\boldsymbol{\mathcal{H}}$ is defined as*

$$\|\boldsymbol{\mathcal{H}}\|_{\circledS\!\!\!\!p} = \left( \sum_{i=1}^{n_3} \left\| \overline{\boldsymbol{\mathcal{H}}}^{(i)} \right\|_{\circledS\!\!\!\!p}^p \right)^{\frac{1}{p}} = \left( \sum_{i=1}^{n_3} \sum_{j=1}^{h} \sigma_j \left( \overline{\boldsymbol{\mathcal{H}}}^{(i)} \right)^p \right)^{\frac{1}{p}}, \tag{1}$$

*where, $0 < p \leqslant 1$, $\sigma_j(\overline{\boldsymbol{\mathcal{H}}}^{(i)})$ denotes the j-th singular value of $\overline{\boldsymbol{\mathcal{H}}}^{(i)}$.*

It should be pointed out that for $0 < p \leqslant 1$ when $p$ is appropriately chosen, the Schatten $p$-norm provides quite effective improvements for a tighter approximation of the rank function [39; 36].

## 4 Methodology

### 4.1 Motivation and Objective

Non-negative matrix factorization (NMF) was initially presented as a dimensionality reduction method, and it is commonly employed as an efficient latent feature learning technique recently. Generally speaking, given a non-negative matrix $\mathbf{X}$, the target of NMF is to decompose $\mathbf{X}$ into two non-negative matrices,

$$\mathbf{X} \approx \mathbf{H}\mathbf{G}^{\mathrm{T}} \tag{2}$$

where $\mathbf{X} \in \mathbb{R}_+^{n \times p}$, $\mathbf{H} \in \mathbb{R}_+^{n \times k}$ and $\mathbf{G} \in \mathbb{R}_+^{p \times k}$. $\mathbb{R}_+^{n \times p}$ means $n$-by-$p$ matrices with elements are all nonnegative. $n$ and $k$ means the number of samples and the number of clusters, respectively.

In order to approximate the matrix before and after factorization, $\ell_2$-norm and F-norm are frequently adopted as the objective function for the NMF. Considering that F-norm can make the model optimization easier, we use F-norm to construct the objective function.

With the extensive use of NMF, more and more variants of NMF have emerged, among which are G-orthogonal NMF [8] and Semi-NMF [9]. By imposing an orthogonality constraint on one of the factor matrices in NMF, we obtain the objective function of the one-side orthogonal NMF,

$$\min_{\mathbf{H} \geqslant 0, \mathbf{G} \geqslant 0} \left\| \mathbf{X} - \mathbf{H}\mathbf{G}^{\mathrm{T}} \right\|_F^2, \quad \text{s.t.} \quad \mathbf{H}^{\mathrm{T}}\mathbf{H} = \mathbf{I}. \tag{3}$$

If we relax the nonnegative constraint on one of the factor matrices in the NMF and the input matrix $\mathbf{X}$ can also be mixed positive and negative, then we can get Semi-NMF. Semi-NMF can be adapted to process input data that has mixed symbols. For G-orthogonal NMF and Semi-NMF, Ding *et al.* [8] presented the following lemma:

**Lemma 1.** *G-orthogonal NMF and Semi-NMF are all relaxation of K-means clustering, and the main advantages of G-orthogonal NMF are (1) Uniqueness of the solution; (2) Excellent clustering interpretability.*

Taking into account the one-side orthogonal NMF, we relax the nonnegative constraints on $\mathbf{X}$ and $\mathbf{G}$. Moreover, inspired by FMCNOF [37], we construct the anchor graph $\mathbf{S}$ obtained from the original data $\mathbf{X}$ as the input of matrix factorization. Compared with the original data, the number of anchors is much smaller, therefore, by adopting the anchor graph constructed by anchors and original data points as the input of matrix factorization, we can reduce the computational complexity of the algorithm effectively.

$$\min_{\mathbf{H} \geqslant 0} \left\| \mathbf{S} - \mathbf{H}\mathbf{G}^{\mathrm{T}} \right\|_F^2, \quad \text{s.t.} \quad \mathbf{H}^{\mathrm{T}}\mathbf{H} = \mathbf{I}, \tag{4}$$

where $\mathbf{S} \in \mathbb{R}^{n \times m}$, $\mathbf{H} \in \mathbb{R}^{n \times k}$ and $\mathbf{G} \in \mathbb{R}^{m \times k}$, $m$ is the number of anchors and we consider $\mathbf{H}$ as the cluster indicator matrix for clustering rows as described in [8]. We will introduce the details of anchor selection and the construction of the anchor graph in the appendix.

As described in the previous section, the existing NMF-based multi-view clustering methods are essentially a matrix factorization on a single view combined with the integration of multiple views. It causes the loss of the original spatial structure of the multi-view data. We extend NMF to the 3rd-order tensor, which can process the multi-view data directly and can also take full advantage of the original spatial structure of the multi-view data. The objective function of tensorial one-side orthogonal non-negative matrix factorization is written in the following form:

$$\min_{\boldsymbol{\mathcal{H}} \geqslant 0} \left\| \boldsymbol{\mathcal{S}} - \boldsymbol{\mathcal{H}} * \boldsymbol{\mathcal{G}}^{\mathrm{T}} \right\|_F^2, \quad \text{s.t.} \quad \boldsymbol{\mathcal{H}}^{\mathrm{T}} * \boldsymbol{\mathcal{H}} = \boldsymbol{\mathcal{I}}, \tag{5}$$

The 3rd-order tensor construction process is illustrated in Fig 2.

In order to better exploit the complementary information and spatial structure between different views, we get inspiration from the excellent performance of the tensor Schatten p-norm [12; 38]. We introduce tensor Schatten p-norm regularization on the tensorial form of the cluster indicator matrix. Our objective function is formulated as follows:

$$\min \left\| \mathcal{S} - \mathcal{H} * \mathcal{G}^{\mathrm{T}} \right\|_F^2 + \lambda \| \mathcal{H} \|_{\circledS\!\!\!\!\!p}^p \qquad s.t. \quad \mathcal{H} \geqslant 0, \mathcal{H}^{\mathrm{T}} * \mathcal{H} = \mathcal{I} \tag{6}$$

where $0 < p \leqslant 1$, $\lambda$ is the hyperparameter of the Schatten $p$-norm term.

**Remark 1.** *The regularizer in the proposed objective (6) is used to explore the complementary information embedded in inter-views cluster assignment matrices $\boldsymbol{H}^{(v)}$ ($v = 1, 2, \cdots, V$). Fig. 2 shows the construction of tensor $\mathcal{H}$, it can be seen that the $k$-th frontal slice $\boldsymbol{\Delta}^{(k)}$ describes the similarity between $N$ sample points and the $k$-th cluster in different views. The idea cluster assignment matrix $\boldsymbol{H}^{(v)}$ should satisfy that the relationship between $N$ data points and the $k$-th cluster is consistent in different views. Since different views usually show different cluster structures, we impose tensor Schatten p-norm minimization [12] constraint on $\mathcal{H}$, which can make sure each $\boldsymbol{\Delta}^{(k)}$ has spatial low-rank structure. Thus $\boldsymbol{\Delta}^{(k)}$ can well characterize the complementary information embedded in inter-views.*

## 4.2 Optimization

Inspired by Augmented Lagrange Multiplier (ALM), we introduce two auxiliary variables $\mathcal{Q}$ and $\mathcal{J}$ and let $\mathcal{H} = \mathcal{Q}$, $\mathcal{H} = \mathcal{J}$, respectively, where $\mathcal{Q} \geqslant 0$. Then, we rewrite the model as the following unconstrained problem:

$$\min \mathcal{L}(\mathcal{Q}, \mathcal{H}, \mathcal{G}, \mathcal{J})$$
$$= \min_{\mathcal{Q} \geqslant 0, \mathcal{H}^{\mathrm{T}} * \mathcal{H} = \mathcal{I}} \left\| \mathcal{S} - \mathcal{H} * \mathcal{G}^{\mathrm{T}} \right\|_F^2 + \lambda \| \mathcal{J} \|_{\circledS\!\!\!\!\!p}^p + \frac{\mu}{2} \left\| \mathcal{H} - \mathcal{Q} + \frac{\mathcal{Y}_1}{\mu} \right\|_F^2 + \frac{\rho}{2} \left\| \mathcal{H} - \mathcal{J} + \frac{\mathcal{Y}_2}{\rho} \right\|_F^2, \tag{7}$$

where $\mathcal{Y}_1$, $\mathcal{Y}_2$ represent Lagrange multipliers and $\mu$, $\rho$ are the penalty parameters. The optimization process can therefore be separated into four steps:

•**Solve $\mathcal{G}$ with fixed $\mathcal{Q}, \mathcal{H}, \mathcal{J}$.** (7) becomes:

$$\min \left\| \mathcal{S} - \mathcal{H} * \mathcal{G}^{\mathrm{T}} \right\|_F^2 \tag{8}$$

After being implemented with discrete Fourier transform (DFT) along the third dimension. the equivalent representation of (8) in the frequency domain becomes:

$$\min \sum_{v=1}^{V} \left\| \overline{\mathcal{S}}^{(v)} - \overline{\mathcal{H}}^{(v)} (\overline{\mathcal{G}}^{(v)})^{\mathrm{T}} \right\|_F^2, \tag{9}$$

where $\overline{\mathcal{G}} = \mathrm{fft}(\mathcal{G}, [\,], 3)$, and the others in the same way.

Let $\Phi = \left\| \overline{\mathcal{S}}^{(v)} - \overline{\mathcal{H}}^{(v)} (\overline{\mathcal{G}}^{(v)})^{\mathrm{T}} \right\|_F^2$, we can obviously get the following equation:

$$\Phi = \mathrm{tr} \left( (\overline{\mathcal{S}}^{(v)})^{\mathrm{T}} \overline{\mathcal{S}}^{(v)} \right) - 2\mathrm{tr} \left( (\overline{\mathcal{H}}^{(v)})^{\mathrm{T}} \overline{\mathcal{S}}^{(v)} \overline{\mathcal{G}}^{(v)} \right) + \mathrm{tr} \left( (\overline{\mathcal{G}}^{(v)})^{\mathrm{T}} \overline{\mathcal{G}}^{(v)} \right). \tag{10}$$

Setting the derivative $\partial \Phi / \partial \overline{\mathcal{G}}^{(v)} = 0$ gives $2\overline{\mathcal{G}}^{(v)} - 2(\overline{\mathcal{S}}^{(v)})^{\mathrm{T}} \overline{\mathcal{H}}^{(v)} = 0$. So the solution of (9) is:

$$\overline{\mathcal{G}}^{(v)} = (\overline{\mathcal{S}}^{(v)})^{\mathrm{T}} \overline{\mathcal{H}}^{(v)} \tag{11}$$

•**Solve $\mathcal{H}$ with fixed $\mathcal{Q}, \mathcal{G}, \mathcal{J}$.** (7) becomes:

$$\min_{\mathcal{H}^{\mathrm{T}} * \mathcal{H} = \mathcal{I}} \left\| \mathcal{S} - \mathcal{H} * \mathcal{G}^{\mathrm{T}} \right\|_F^2 + \frac{\mu}{2} \left\| \mathcal{H} - \mathcal{Q} + \frac{\mathcal{Y}_1}{\mu} \right\|_F^2 + \frac{\rho}{2} \left\| \mathcal{H} - \mathcal{J} + \frac{\mathcal{Y}_2}{\rho} \right\|_F^2 \tag{12}$$

And (12) is equivalent to the following in the frequency domain:

$$
\min_{(\overline{\mathcal{H}}^{(v)})^{\mathrm{T}}\overline{\mathcal{H}}^{(v)}=\mathbf{I}} \sum_{v=1}^{V} \left\| \overline{\mathcal{S}}^{(v)} - \overline{\mathcal{H}}^{(v)}(\overline{\mathcal{G}}^{(v)})^{\mathrm{T}} \right\|_F^2
$$
$$
+ \sum_{v=1}^{V} \frac{\mu}{2} \left\| \overline{\mathcal{H}}^{(v)} - \overline{\mathcal{Q}}^{(v)} + \frac{\overline{\mathcal{Y}}_1^{(v)}}{\mu} \right\|_F^2 + \sum_{v=1}^{V} \frac{\rho}{2} \left\| \overline{\mathcal{H}}^{(v)} - \overline{\mathcal{J}}^{(v)} + \frac{\overline{\mathcal{Y}}_2^{(v)}}{\rho} \right\|_F^2, \tag{13}
$$

where $\overline{\mathcal{H}} = \mathrm{fft}(\mathcal{H}, [\,], 3)$, and the others in the same way.

And (13) can be reduced to:

$$
\min_{(\overline{\mathcal{H}}^{(v)})^{\mathrm{T}}\overline{\mathcal{H}}^{(v)}=\mathbf{I}} -2\mathrm{tr}\left( \overline{\mathcal{G}}^{(v)}(\overline{\mathcal{H}}^{(v)})^{\mathrm{T}}\overline{\mathcal{S}}^{(v)} \right) - \mu\mathrm{tr}\left( (\overline{\mathcal{H}}^{(v)})^{\mathrm{T}}\overline{\mathcal{W}}_1^{(v)} \right) - \rho\mathrm{tr}\left( (\overline{\mathcal{H}}^{(v)})^{\mathrm{T}}\overline{\mathcal{W}}_2^{(v)} \right) \tag{14}
$$

where $\overline{\mathcal{W}}_1^{(v)} = \overline{\mathcal{Q}}^{(v)} - \frac{\overline{\mathcal{Y}}_1^{(v)}}{\mu}$ and $\overline{\mathcal{W}}_2^{(v)} = \overline{\mathcal{J}}^{(v)} - \frac{\overline{\mathcal{Y}}_2^{(v)}}{\rho}$.

and it also can be reduced to:

$$
\max_{(\overline{\mathcal{H}}^{(v)})^{\mathrm{T}}\overline{\mathcal{H}}^{(v)}=\mathbf{I}} \mathrm{tr}\left( (\overline{\mathcal{H}}^{(v)})^{\mathrm{T}}\overline{\mathcal{B}}^{(v)} \right) \tag{15}
$$

where $\overline{\mathcal{B}}^{(v)} = 2\overline{\mathcal{S}}^{(v)}\overline{\mathcal{G}}^{(v)} + \mu\overline{\mathcal{W}}_1^{(v)} + \rho\overline{\mathcal{W}}_2^{(v)}$.

To solve (15), we introduce the following Theorem:

**Theorem 1.** *Given* $\mathbf{G}$ *and* $\mathbf{P}$, *where* $\mathbf{G}(\mathbf{G})^{\mathrm{T}} = \mathbf{I}$ *and* $\mathbf{P}$ *has the singular value decomposition* $\mathbf{P} = \mathbf{\Lambda}\mathbf{S}(\mathbf{V})^{\mathrm{T}}$, *then the optimal solution of*

$$
\max_{\mathbf{G}(\mathbf{G})^{\mathrm{T}}=\mathbf{I}} \mathrm{tr}(\mathbf{G}\mathbf{P}) \tag{16}
$$

*is* $\mathbf{G}^* = \mathbf{V}[\mathbf{I}, \mathbf{0}](\mathbf{\Lambda})^{\mathrm{T}}$.

*Proof.* From the SVD $\mathbf{P} = \mathbf{\Lambda}\mathbf{S}(\mathbf{V})^{\mathrm{T}}$ and together with (16), it is evident that

$$
\mathrm{tr}(\mathbf{G}\mathbf{P}) = \mathrm{tr}(\mathbf{G}\mathbf{\Lambda}\mathbf{S}(\mathbf{V})^{\mathrm{T}}) = \mathrm{tr}(\mathbf{S}(\mathbf{V})^{\mathrm{T}}\mathbf{G}\mathbf{\Lambda}) = \mathrm{tr}(\mathbf{S}\mathbf{H}) = \sum_i s_{ii}h_{ii}, \tag{17}
$$

where $\mathbf{H} = (\mathbf{V})^{\mathrm{T}}\mathbf{G}\mathbf{\Lambda}$, $s_{ii}$ and $h_{ii}$ are the $(i, i)$ elements of $\mathbf{S}$ and $\mathbf{H}$, respectively. It can be easily verified that $\mathbf{H}(\mathbf{H})^{\mathrm{T}} = \mathbf{I}$, where $\mathbf{I}$ is an identity matrix. Therefore $-1 \leqslant h_{ii} \leqslant 1$ and $s_{ii} \geqslant 0$, Thus we have:

$$
\mathrm{tr}(\mathbf{G}\mathbf{P}) = \sum_i s_{ii}h_{ii} \leqslant \sum_i s_{ii}. \tag{18}
$$

The equality holds when $\mathbf{H}$ is an identity matrix. $\mathrm{tr}(\mathbf{G}\mathbf{P})$ reaches the maximum when $\mathbf{H} = [\mathbf{I}, \mathbf{0}]$. $\quad\square$

So the solution of (15) is:

$$
\overline{\mathcal{H}}^{(v)} = \overline{\mathbf{\Lambda}}^{(v)}(\overline{\mathbf{V}}^{(v)})^{\mathrm{T}} \tag{19}
$$

where $\overline{\mathbf{\Lambda}}^{(v)}$ and $\overline{\mathbf{V}}^{(v)}$ can be obtained by SVD $\overline{\mathcal{B}}^{(v)} = \overline{\mathbf{\Lambda}}^{(v)}\mathbf{X}(\overline{\mathbf{V}}^{(v)})^{\mathrm{T}}$

●**Solve** $\mathcal{Q}$ **with fixed** $\mathcal{H}, \mathcal{G}, \mathcal{J}$. (7) becomes:

$$
\min_{\mathcal{Q}\geqslant 0} \frac{\mu}{2} \left\| \mathcal{H} - \mathcal{Q} + \frac{\mathcal{Y}_1}{\mu} \right\|_F^2 \tag{20}
$$

(20) is obviously equivalent to:

$$
\min_{\mathcal{Q}\geqslant 0} \frac{\mu}{2} \left\| \mathcal{Q} - (\mathcal{H} + \frac{\mathcal{Y}_1}{\mu}) \right\|_F^2 \tag{21}
$$

According to [37], the solution of (21) is:

$$\mathcal{Q} = \left(\mathcal{H} + \frac{\mathcal{Y}_1}{\mu}\right)_+ \tag{22}$$

•**Solve $\mathcal{J}$ with fixed $\mathcal{Q}, \mathcal{H}, \mathcal{G}$.** (7) becomes:

$$\min \lambda \|\mathcal{J}\|_{\circledS\!\!\!\!P}^p + \frac{\rho}{2} \left\| \mathcal{H} - \mathcal{J} + \frac{\mathcal{Y}_2}{\rho} \right\|_F^2, \tag{23}$$

after completing the square regarding $\mathcal{J}$, we can deduce

$$\mathcal{J}^* = \arg\min \frac{1}{2} \left\| \mathcal{H} + \frac{\mathcal{Y}_2}{\rho} - \mathcal{J} \right\|_F^2 + \frac{\lambda}{\rho} \|\mathcal{J}\|_{\circledS\!\!\!\!P}^p, \tag{24}$$

which has a closed-form solution as Lemma 2 [12]:

**Lemma 2.** *Let $\mathcal{Z} \in \mathbb{R}^{n_1 \times n_2 \times n_3}$ have a t-SVD $\mathcal{Z} = \mathcal{U} * \mathcal{S} * \mathcal{V}^{\mathrm{T}}$, then the optimal solution for*

$$\min_{\mathcal{X}} \tfrac{1}{2} \|\mathcal{X} - \mathcal{Z}\|_F^2 + \tau \|\mathcal{X}\|_{\circledS\!\!\!\!P}^p. \tag{25}$$

*is $\mathcal{X}^* = \Gamma_\tau(\mathcal{Z}) = \mathcal{U} * \mathrm{ifft}(P_\tau(\overline{\mathcal{Z}})) * \mathcal{V}^{\mathrm{T}}$, where $P_\tau(\overline{\mathcal{Z}})$ is an f-diagonal 3rd-order tensor, whose diagonal elements can be found by using the GST algorithm introduced in [12].*

Now the solution of (24) is:

$$\mathcal{J}^* = \Gamma_{\frac{\lambda}{\rho}}(\mathcal{H} + \frac{\mathcal{Y}_2}{\rho}). \tag{26}$$

Finally, the optimization procedure for Multi-View Clustering via Orthogonal non-negative Tensor Factorization (Orth-NTF) is outlined in Algorithm 1.

---

**Algorithm 1** Multi-View Clustering via Orthogonal non-negative Tensor Factorization (Orth-NTF)

---

**Input**: Data matrices $\{\mathbf{X}^{(v)}\}_{v=1}^V \in \mathbb{R}^{N \times d_v}$; anchors numbers $m$; cluster number $K$.
**Output**: Cluster labels $\mathbf{Y}$ of each data points.
**Initialize**: $\mu = 10^{-5}, \rho = 10^{-5}, \eta = 1.6, \mathcal{Y}_1 = 0, \mathcal{Y}_2 = 0, \overline{\mathbf{Q}}^{(v)}$ is identity matrix;
1: Compute graph matrix $\mathbf{S}^{(v)}$ of each views;
2: **while** not condition **do**
3:    Update $\overline{\mathcal{G}}^{(v)}$ by solving (11);
4:    Update $\overline{\mathcal{H}}^{(v)}$ by solving (19);
5:    Update $\overline{\mathcal{Q}}^{(v)}$ by solving (22);
6:    Update $\mathcal{J}$ by using (24);
7:    Update $\mathcal{Y}_1, \mathcal{Y}_2, \mu$ and $\rho$: $\mathcal{Y}_1 = \mathcal{Y}_1 + \mu(\mathcal{H} - \mathcal{Q}), \mathcal{Y}_2 = \mathcal{Y}_2 + \mu(\mathcal{H} - \mathcal{J}), \mu = \min(\eta\mu, 10^{13})$, $\rho = \min(\eta\rho, 10^{13})$;
8: **end while**
9: Calculate the $K$ clusters by using
    $\mathbf{H} = \sum_{v=1}^V \mathbf{H}^{(v)}/V$;
10: **return** Clustering result (The position of the largest element in each row of the indicator matrix is the label of the corresponding sample).

---

### 4.3 Convergence Analysis

**Theorem 2.** *[Convergence Analysis of Algorithm 1] Let $\mathcal{P}_k = \{\mathcal{Q}_k, \mathcal{H}_k, \mathcal{G}_k, \mathcal{J}_k, \mathcal{Y}_{2,k}, \mathcal{Y}_{1,k}\}, 1 \le k < \infty$ in (7) be a sequence generated by **Algorithm 1**, then*

1. *$\mathcal{P}_k$ is bounded with the assumption $\lim_{k \to 0} \max\{\mu_k, \rho_k\}(\overline{\mathcal{H}}_{k+1}^{(v)} - \overline{\mathcal{H}}_k^{(v)}) = 0$;*

2. *Any accumulation point of $\mathcal{P}_k$ is a stationary KKT point of (7).*

The proof will be provided in the appendix and we need to mention that the KKT conditions can be used to determine the stop conditions for Algorithm 1, which are $\|\mathcal{Q}_k - \mathcal{H}_k\|_\infty \le \varepsilon$, $\|\mathcal{Q}_k - \mathcal{J}_k\|_\infty \le \varepsilon$.

Table 1: Multi-view datasets used in our experiments

| #Dataset | #Samples | #View | #Class | #Feature |
|---|---|---|---|---|
| MSRC | 210 | 5 | 7 | 24, 576, 512, 256, 254 |
| HandWritten4 | 2000 | 4 | 10 | 76, 216, 47, 6 |
| Mnist4 | 4000 | 3 | 4 | 30, 9, 30 |
| Reuters | 18758 | 5 | 6 | 21531, 24892, 34251, 15506, 11547 |
| Noisy MNIST | 50000 | 2 | 10 | 784, 784 |

### 4.4 Complexity Analysis

For Orth-NTF, the storage requirements for $\mathcal{G}$, $\mathcal{H}$, $\mathcal{Q}$, $\mathcal{J}$, $\mathcal{Y}_1$ and $\mathcal{Y}_2$ have complexities of $\mathcal{O}(V(m+k)n)$, $\mathcal{O}(V(n+k)k)$, $\mathcal{O}(Vnk)$, $\mathcal{O}(Vnk)$, $\mathcal{O}(Vnk)$ and $\mathcal{O}(Vnk)$, respectively. Combining these, the total storage complexity for Orth-NTF is $\mathcal{O}(Vnm + vk^2 + 6Vnk)$.

For the computational complexity, the process of constructing $\mathcal{S}$ has a computational complexity of $\mathcal{O}(Vnmd + Vnm\log(m))$. When updating the four variables, $\mathcal{G}$, $\mathcal{H}$, $\mathcal{Q}$ and $\mathcal{J}$, their respective computational complexities are $\mathcal{O}(Vnmd + Vnm\log(m))$, $\mathcal{O}(Vm^2k + Vmk^2)$, $\mathcal{O}(Vnk)$ and $\mathcal{O}(2Vnk\log(Vk) + V^2kn)$. Given that $m$, $n$, $k$ and $V$ are relatively small constants, the primary computational cost associated with updating the variables stands at $\mathcal{O}(Vnkm + Vm^2k)$. Summing it all up, the overall computational complexity of our proposed method is $\mathcal{O}(Vnmd + Vm^2k)$.

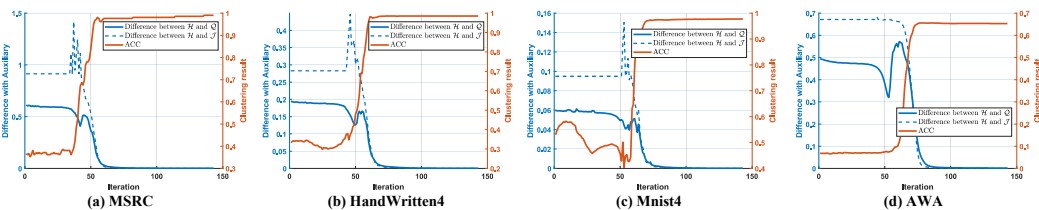

(a) MSRC      (b) HandWritten4      (c) Mnist4      (d) AWA

Figure 3: Convergence experiments on MSRC, HandWritten4, Mnist4 and AWA.

## 5 Experiments

In this section, we demonstrate the performance of our proposed method through extensive experiments. It is compared with plenty of state-of-art multi-view clustering algorithms on some multi-view datasets. We evaluate the clustering performance by applying 7 metrics used widely, *i.e.*, 1) ACC; 2) NMI; 3) Purity; 4) PRE; 5) REC; 6) F-score; and 7) ARI. The higher the value the better the clustering results for all metrics mentioned above. Detailed experimental configurations and hyper-parameters on each dataset are in the appendix.

### 5.1 Datasets and Compared Baselines Methods

The following multi-view datasets are selected to examine our proposed method. The details of the datasets are shown in Table 1. **MSRC** [34]; **HandWritten4** [10]; **Mnist4** [6]; **AWA** [11]; **Reuters** [2]; **Noisy MNIST** [32]; We choose the following 8 state-of-art multi-view clustering algorithms to compare with our proposed methods: **AMGL** [25]; **MVGL** [40]; **CSMSC** [24]; **GMC** [30]; **LMVSC** [17]; **SMSC** [16]; **SFMC** [21] **FMCNOF**[37]; **FPMVS-CAG** [31]; **ETLMSC** [35]; **MSC-BG** [38];

### 5.2 Experiments Result

The clustering performances are listed in Table 2 and Table 3. They contain four medium-scale datasets and two large-scale datasets. The corresponding experimental configurations and descriptions are included in the appendix. It is clear that our algorithm outperforms the other baseline algorithms on most of the datasets.

This advantage may stem from the fact that our model directly factorizes the tensorized anchor graph—comprised of anchor graphs from various views—into the product of two non-negative tensors, one being an index tensor. As a result, our model effectively captures both the spatial structural information and the complementary data present in the anchor graphs from different

Table 2: Clustering performance on MSRC, HandWritten4, Mnist4 and AWA. (The best result is in bold, and the second-best result is underlined.)

| Dataset | MSRC | | | | | | |
|---|---|---|---|---|---|---|---|
| Metrics | ACC | NMI | Purity | PER | REC | F-score | ARI |
| AMGL [25] (IJCAI' 16) | 0.751±0.000 | 0.704±0.000 | 0.789±0.000 | 0.621±0.000 | 0.744±0.000 | 0.674±0.000 | 0.615±0.000 |
| MVGL [40] (TCYB' 17) | 0.690±0.000 | 0.663±0.000 | 0.733±0.000 | 0.466±0.000 | 0.715±0.000 | 0.564±0.000 | 0.476±0.000 |
| CSMSC [24] (AAAI' 18) | 0.758±0.007 | 0.735±0.010 | 0.793±0.008 | 0.736±0.014 | 0.673±0.008 | 0.703±0.010 | 0.653±0.012 |
| GMC [30] (TKDE' 19) | 0.895±0.000 | 0.809±0.000 | 0.895±0.000 | 0.788±0.000 | 0.814±0.000 | 0.801±0.000 | 0.768±0.000 |
| LMVSC [17] (AAAI' 20) | 0.814±0.000 | 0.717±0.000 | 0.814±0.000 | 0.676±0.000 | 0.692±0.000 | 0.684±0.000 | 0.632±0.000 |
| SMSC [16] (Inf Fusion' 20) | 0.766±0.000 | 0.717±0.000 | 0.804±0.000 | 0.672±0.000 | 0.718±0.000 | 0.694±0.000 | 0.643±0.000 |
| FMCNOF [37] (TIP' 21) | 0.440±0.039 | 0.345±0.046 | 0.449±0.042 | 0.290±0.036 | 0.606±0.074 | 0.395±0.036 | 0.249±0.051 |
| FPMVS-CAG [31] (TIP' 21) | 0.786±0.000 | 0.686±0.000 | 0.786±0.000 | 0.684±0.000 | 0.642±0.000 | 0.731±0.000 | 0.629±0.000 |
| SFMC [21] (TPAMI' 22) | 0.810±0.000 | 0.721±0.000 | 0.810±0.000 | 0.657±0.000 | 0.782±0.000 | 0.714±0.000 | 0.663±0.000 |
| ETLMSC [35] (TIP' 19) | 0.962±0.000 | 0.937±0.000 | 0.962±0.000 | 0.926±0.000 | 0.931±0.000 | 0.928±0.000 | 0.917±0.000 |
| MSC-BG [38] (TIP' 22) | 0.981±0.000 | 0.960±0.000 | 0.981±0.000 | 0.961±0.000 | 0.963±0.000 | 0.962±0.000 | 0.956±0.000 |
| ours | **0.990±0.000** | **0.978±0.000** | **0.990±0.000** | **0.980±0.000** | **0.981±0.000** | **0.981±0.000** | **0.978±0.000** |

| Dataset | HandWritten4 | | | | | | |
|---|---|---|---|---|---|---|---|
| Metrics | ACC | NMI | Purity | PER | REC | F-score | ARI |
| AMGL [25] (IJCAI' 16) | 0.704±0.000 | 0.762±0.000 | 0.732±0.000 | 0.591±0.000 | 0.781±0.000 | 0.670±0.000 | 0.628±0.000 |
| MVGL [40] (TCYB' 17) | 0.811±0.000 | 0.809±0.000 | 0.831±0.000 | 0.721±0.000 | 0.826±0.000 | 0.770±0.000 | 0.743±0.000 |
| CSMSC [24] (AAAI' 18) | 0.806±0.001 | 0.793±0.001 | 0.867±0.001 | 0.778±0.001 | 0.743±0.001 | 0.760±0.001 | 0.733±0.001 |
| GMC [30] (TKDE' 19) | 0.861±0.000 | 0.859±0.000 | 0.861±0.000 | 0.799±0.000 | 0.855±0.000 | 0.826±0.000 | 0.806±0.000 |
| LMVSC [17] (AAAI' 20) | 0.904±0.000 | 0.831±0.000 | 0.904±0.000 | 0.819±0.000 | 0.825±0.000 | 0.822±0.000 | 0.802±0.000 |
| SMSC [16] (Inf Fusion' 20) | 0.742±0.000 | 0.781±0.000 | 0.759±0.000 | 0.675±0.000 | 0.767±0.000 | 0.717±0.000 | 0.685±0.000 |
| FMCNOF [37] (TIP' 21) | 0.385±0.092 | 0.370±0.092 | 0.386±0.093 | 0.254±0.077 | 0.688±0.101 | 0.360±0.070 | 0.250±0.097 |
| FPMVS-CAG [31] (TIP' 21) | 0.744±0.000 | 0.753±0.000 | 0.744±0.000 | 0.681±0.000 | 0.636±0.000 | 0.762±0.000 | 0.642±0.000 |
| SFMC [21] (TPAMI' 22) | 0.853±0.000 | 0.871±0.000 | 0.873±0.000 | 0.775±0.000 | _0.910±0.000_ | 0.837±0.000 | 0.817±0.000 |
| ETLMSC [35] (TIP' 19) | _0.938±0.001_ | 0.893±0.001 | _0.938±0.001_ | 0.886±0.001 | 0.890±0.001 | 0.888±0.001 | 0.876±0.001 |
| MSC-BG [38] (TIP' 22) | 0.889±0.000 | 0.922±0.000 | 0.889±0.000 | 0.871±0.000 | 0.893±0.000 | 0.882±0.000 | 0.869±0.000 |
| ours | **0.985±0.000** | **0.969±0.000** | **0.985±0.000** | **0.970±0.000** | **0.970±0.000** | **0.970±0.000** | **0.966±0.000** |

| Dataset | Mnist4 | | | | | | |
|---|---|---|---|---|---|---|---|
| Metrics | ACC | NMI | Purity | PER | REC | F-score | ARI |
| AMGL [25] (IJCAI' 16) | 0.921±0.000 | 0.806±0.000 | 0.921±0.000 | 0.854±0.000 | 0.862±0.000 | 0.858±0.000 | 0.810±0.000 |
| MVGL [40] (TCYB' 17) | 0.919±0.000 | 0.803±0.000 | 0.919±0.000 | 0.851±0.000 | 0.860±0.000 | 0.856±0.000 | 0.807±0.000 |
| CSMSC [24] (AAAI' 18) | 0.641±0.000 | 0.601±0.010 | 0.728±0.008 | 0.607±0.014 | 0.767±0.008 | 0.677±0.010 | 0.553±0.012 |
| GMC [30] (TKDE' 19) | 0.920±0.000 | 0.807±0.000 | 0.920±0.000 | 0.853±0.000 | 0.861±0.000 | 0.857±0.000 | 0.809±0.000 |
| LMVSC [17] (AAAI' 20) | 0.892±0.000 | 0.726±0.000 | 0.892±0.000 | 0.808±0.000 | 0.812±0.000 | 0.810±0.000 | 0.747±0.000 |
| SMSC [16] (Inf Fusion' 20) | 0.909±0.000 | 0.774±0.000 | 0.909±0.000 | 0.834±0.000 | 0.841±0.000 | 0.837±0.000 | 0.783±0.000 |
| FMCNOF [37] (TIP' 21) | 0.697±0.119 | 0.490±0.102 | 0.711±0.096 | 0.558±0.118 | 0.683±0.073 | 0.611±0.095 | 0.460±0.145 |
| FPMVS-CAG [31] (TIP' 21) | 0.885±0.000 | 0.715±0.000 | 0.885±0.000 | 0.800±0.000 | 0.795±0.000 | 0.815±0.000 | 0.733±0.000 |
| SFMC [21] (TPAMI' 22) | 0.916±0.000 | 0.797±0.000 | 0.916±0.000 | 0.846±0.000 | 0.855±0.000 | 0.850±0.000 | 0.800±0.000 |
| ETLMSC [35] (TIP' 19) | 0.934±0.000 | 0.847±0.000 | 0.934±0.000 | 0.878±0.000 | 0.885±0.000 | 0.881±0.000 | 0.842±0.000 |
| MSC-BG [38] (TIP' 22) | _0.938±0.000_ | _0.861±0.000_ | _0.938±0.000_ | _0.884±0.000_ | _0.891±0.000_ | _0.888±0.000_ | _0.850±0.000_ |
| ours | **0.977±0.000** | **0.926±0.000** | **0.977±0.000** | **0.955±0.000** | **0.956±0.000** | **0.955±0.000** | **0.941±0.000** |

| Dataset | AWA | | | | | | |
|---|---|---|---|---|---|---|---|
| Metrics | ACC | NMI | Purity | PER | REC | F-score | ARI |
| MVGL [40] (TCYB' 17) | 0.061±0.000 | 0.070±0.000 | 0.065±0.000 | 0.020±0.000 | _0.843±0.000_ | 0.040±0.000 | 0.002±0.000 |
| CSMSC [24] (AAAI' 18) | 0.113±0.000 | 0.175±0.002 | 0.119±0.001 | 0.051±0.001 | 0.054±0.000 | 0.053±0.001 | 0.033±0.001 |
| GMC [30] (TKDE' 19) | 0.028±0.000 | 0.030±0.000 | 0.039±0.000 | 0.020±0.000 | **0.915±0.000** | 0.039±0.000 | 0.001±0.000 |
| LMVSC [17] (AAAI' 20) | 0.105±0.000 | 0.171±0.000 | 0.114±0.000 | 0.041±0.000 | 0.065±0.000 | 0.051±0.000 | 0.027±0.000 |
| FMCNOF [37] (TIP' 21) | 0.035±0.008 | 0.018±0.011 | 0.035±0.008 | 0.021±0.001 | 0.688±0.227 | 0.040±0.002 | 0.002±0.003 |
| FPMVS-CAG [31] (TIP' 21) | 0.106±0.000 | 0.183±0.000 | 0.111±0.000 | 0.069±0.000 | 0.047±0.000 | 0.142±0.000 | 0.042±0.000 |
| SFMC [21] (TPAMI' 22) | 0.042±0.000 | 0.044±0.000 | 0.049±0.000 | 0.023±0.000 | 0.592±0.000 | 0.044±0.000 | 0.006±0.000 |
| ETLMSC [35] (TIP' 19) | _0.631±0.000_ | _0.783±0.000_ | _0.656±0.000_ | _0.498±0.000_ | 0.580±0.000 | _0.536±0.000_ | _0.526±0.000_ |
| MSC-BG [38] (TIP' 22) | 0.493±0.000 | 0.550±0.000 | 0.511±0.000 | 0.059±0.000 | 0.494±0.000 | 0.105±0.000 | 0.073±0.000 |
| ours | **0.646±0.000** | **0.815±0.000** | **0.670±0.000** | **0.534±0.000** | 0.574±0.000 | **0.553±0.000** | **0.543±0.000** |

perspectives. Additionally, with orthogonal and non-negative constraints in place, our model offers clear interpretability for clustering. This means that each row of the indicator matrix for every view contains a single non-zero element, with its position indicating the label of the associated sample. Consequently, our model can immediately provide the label without necessitating any post-processing, a step which other methods still require.

### 5.3 Ablation Experiments

We do some ablation experiments on orthogonal constraint and Schatten p-norm on four datasets as shown in Table 4. It can be found that, tensor Schatten p-norm regularization is overall superior to orthogonal constraint. The reason is that tensor Schatten p-norm regularization effectively characterizes both the complementary information and spatial structure information of index matrices of different views. Compared to orthogonal and tensor Schatten p-norm constraints, Joint constraints have great contribution for clustering.

### 5.4 Experiments of convergence

We optimize the objective function iteratively by introducing two auxiliary variables $\mathcal{Q}$ and $\mathcal{J}$. We test the convergence of our algorithm by checking the difference between $\mathcal{H}$ and the two auxiliary

Table 3: Clustering results and running time (sec.) on Reuters and NoisyMNIST. ("OM" means out of memory and "-" means the algorithm takes more than three hours to calculate.)

| Dataset | Reuters | | | | Dataset | Noisy MNIST | | | |
|---|---|---|---|---|---|---|---|---|---|
| Metrics | ACC | NMI | Purity | Time | Metrics | ACC | NMI | Purity | Time |
| AMGL [25] (IJCAI' 16) | OM | OM | OM | OM | AMGL [25] (IJCAI' 16) | OM | OM | OM | OM |
| MVGL [40] (TCYB' 17) | OM | OM | OM | OM | MVGL [40] (TCYB' 17) | OM | OM | OM | OM |
| CSMSC [24] (AAAI' 18) | OM | OM | OM | OM | CSMSC [24] (AAAI' 18) | OM | OM | OM | OM |
| GMC [30] (TKDE' 19) | - | - | - | - | GMC [30] (TKDE' 19) | - | - | - | - |
| LMVSC [17] (AAAI' 20) | 0.587±0.000 | 0.335±0.000 | 0.616±0.000 | **150.51** | LMVSC [17] (AAAI' 20) | 0.388±0.000 | 0.344±0.000 | 0.434±0.000 | **151.14** |
| SMSC [16] (Inf Fusion' 20) | OM | OM | OM | OM | SMSC [16] (Inf Fusion' 20) | OM | OM | OM | OM |
| FMCNOF [37] (TIP' 21) | 0.343±0.007 | 0.125±0.037 | 0.358±0.052 | 186.45 | FMCNOF [37] (TIP' 21) | 0.333±0.038 | 0.237±0.032 | 0.340±0.032 | 192.60 |
| FPMVS-CAG [31] (TIP' 21) | 0.576±0.000 | 0.359±0.000 | 0.637±0.000 | 2252.20 | FPMVS-CAG [31] (TIP' 21) | 0.554±0.000 | 0.513±0.000 | 0.567±0.000 | 2258.06 |
| SFMC [21] (TPAMI' 22) | 0.602±0.000 | 0.354±0.000 | 0.552±0.000 | 494.68 | SFMC [21] (TPAMI' 22) | 0.699±0.000 | 0.681±0.000 | 0.727±0.000 | 495.90 |
| ETLMSC [35] (TIP' 19) | OM | OM | OM | OM | ETLMSC [35] (TIP' 19) | OM | OM | OM | OM |
| MSC-BG [38] (TIP'22) | 0.640±0.000 | 0.484±0.000 | 0.686±0.000 | 462.33 | MSC-BG [38] (TIP'22) | OM | OM | OM | OM |
| ours | **0.694±0.000** | **0.686±0.000** | **0.809±0.000** | 557.09 | ours | **0.701±0.000** | **0.729±0.000** | **0.747±0.000** | 563.66 |

Table 4: Ablation experiments on MSRC, HandWritten4, Mnist4 and AWA. ($\mathcal{C}_{orth}$ and $\mathcal{C}_{Sp}$ represent the orthogonal constraint and the tensor Schatten $p$-norm regularization, respectively.)

| Situations | | MSRC | | | HandWritten4 | | |
|---|---|---|---|---|---|---|---|
| $\mathcal{C}_{orth}$ | $\mathcal{C}_{Sp}$ | ACC | NMI | Purity | ACC | NMI | Purity |
| ✗ | ✗ | 0.776 | 0.653 | 0.776 | 0.559 | 0.572 | 0.581 |
| ✔ | ✗ | 0.785 | 0.659 | 0.785 | 0.594 | 0.621 | 0.615 |
| ✗ | ✔ | 0.886 | 0.819 | 0.886 | 0.725 | 0.741 | 0.725 |
| ✔ | ✔ | 0.990 | 0.987 | 0.990 | 0.985 | 0.969 | 0.985 |

| Situations | | Mnist4 | | | AWA | | |
|---|---|---|---|---|---|---|---|
| $\mathcal{C}_{orth}$ | $\mathcal{C}_{Sp}$ | ACC | NMI | Purity | ACC | NMI | Purity |
| ✗ | ✗ | 0.898 | 0.750 | 0.898 | 0.020 | 0.020 | 0.020 |
| ✔ | ✗ | 0.905 | 0.759 | 0.905 | 0.092 | 0.147 | 0.097 |
| ✗ | ✔ | 0.912 | 0.789 | 0.912 | 0.212 | 0.125 | 0.213 |
| ✔ | ✔ | 0.977 | 0.926 | 0.977 | 0.646 | 0.815 | 0.670 |

variables. The result of the experiment is shown in Fig 7. It is evident that when the iteration reaches around 80, the difference decreases significantly until it is about zero finally. When the number of iteration increases, the clustering metric (such as ACC) overall improves gradually and tends to be constant with the convergence of the algorithm. It also indicates that our method has good clustering performances.

**More experiments can be found in the appendix.**

# 6   Conclusion

We are concerned in this paper with multi-view clustering based on semi-non-negative tensor factorization (Orth-NTF) with one-side orthogonal constraint. Our proposed model extends NMF to Orth-NTF so that the spatial structure information of the multi-view data can be utilized to improve the clustering performance. In addition, the complementary information embedded in different views is fully leveraged by imposing the tensor Schatten $p$-norm composed of cluster indicator matrices. To diminish the computational complexity, we adopt anchor graphs instead of the original multi-view data. Also, we provide an optimization algorithm for the proposed method and validate the effectiveness of this approach in extensive experiments on different datasets.

# Acknowledgements

The authors would like to thank the anonymous reviewers, AC and PC for their constructive comments and suggestions. This work was supported in part by National NSFC under Grants 62176203, in part by the Natural Science Foundation of Shandong Province under Grant ZR202102180986, in part by the State Key Laboratory of Multimodal Artificial Intelligence Systems under Grant 202200035, in part by Guangxi Key Laboratory of Digital Infrastructure under Grant GXDIOP2023010, in part by the Fundamental Research Funds for the Central Universities, the Innovation Fund of Xidian University.

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

# A   Proof of Convergence

## A.1   Proof of the 1st part

**Lemma 3** (Proposition 6.2 of [20]). *Suppose $F : \mathbb{R}^{n_1 \times n_2} \to \mathbb{R}$ is represented as $F(X) = f \circ \sigma(X)$, where $X \in \mathbb{R}^{n_1 \times n_2}$ with SVD $X = U \mathrm{diag}(\sigma_1, \ldots, \sigma_n) V^{\mathrm{T}}$, $n = \min(n_1, n_2)$, and $f$ is differentiable. The gradient of $F(X)$ at $X$ is*

$$\frac{\partial F(X)}{\partial X} = U \mathrm{diag}(\theta) V^{\mathrm{T}}, \tag{27}$$

*where $\theta = \frac{\partial f(y)}{\partial y}\big|_{y=\sigma(X)}$.*

To minimize $\bar{\mathcal{Q}}^{(v)}$ at step $k+1$ in (21) in the main body, the optimal $\bar{\mathcal{Q}}_{k+1}^{(v)}$ needs to satisfy the first-order optimal condition

$$\bar{\mathcal{Q}}_{k+1}^{(v)} = \bar{\mathcal{H}}_k^{(v)} + \frac{\bar{\mathcal{Y}}_{1,k}}{\mu_k}.$$

By using the updating rule $\bar{\mathcal{Y}}_{1,k+1}^{(v)} = \bar{\mathcal{Y}}_{1,k}^{(v)} + \mu_k (\bar{\mathcal{H}}_k^{(v)} - \bar{\mathcal{Q}}_k^{(v)})$, we have

$$\frac{\bar{\mathcal{Y}}_{1,k+1}^{(v)}}{\mu_k} + (\bar{\mathcal{Q}}_k^{(v)} - \bar{\mathcal{Q}}_{k+1}^{(v)}) = 0.$$

According to our assumption $\lim_{k \to 0} \mu_k (\bar{\mathcal{Q}}_{k+1}^{(v)} - \bar{\mathcal{Q}}_k^{(v)}) = 0$, we know $\mathcal{Y}_{1,k+1}$ is bounded.

To minimize $\mathcal{J}$ at step $k+1$ in (24) in the main body, the optimal $\mathcal{J}_{k+1}$ needs to satisfy the first-order optimal condition

$$\lambda \nabla_{\mathcal{J}} \|\mathcal{J}_{k+1}\|_{\circledS \mathfrak{p}}^p + \rho_k (\mathcal{J}_{k+1} - \mathcal{H}_{k+1} - \frac{1}{\rho_k} \mathcal{Y}_{2,k}) = 0.$$

Recall that when $0 < p < 1$, in order to overcome the singularity of $(|\eta|^p)' = p\eta/|\eta|^{2-p}$ near $\eta = 0$, we consider for $0 < \epsilon \ll 1$ the approximation

$$\partial |\eta|^p \approx \frac{p\eta}{\max\{\epsilon^{2-p}, |\eta|^{2-p}\}}.$$

Letting $\overline{\mathcal{J}}^{(i)} = \overline{\mathcal{U}}^{(i)} \mathrm{diag}\left(\sigma_j(\overline{\mathcal{J}}^{(i)})\right) \overline{\mathcal{V}}^{(i)\mathrm{H}}$, then it follows from Lemma 3 that

$$\frac{\partial \|\overline{\mathcal{J}}^{(i)}\|_{\circledS \mathfrak{p}}^p}{\partial \overline{\mathcal{J}}^{(i)}} = \overline{\mathcal{U}}^{(i)} \mathrm{diag}\left(\frac{p\sigma_j(\overline{\mathcal{J}}^{(i)})}{\max\{\epsilon^{2-p}, |\sigma_j(\overline{\mathcal{J}}^{(i)})|^{2-p}\}}\right) \overline{\mathcal{V}}^{(i)\mathrm{H}}.$$

And then one can obtain

$$\frac{p\sigma_j(\overline{\mathcal{J}}^{(i)})}{\max\{\epsilon^{2-p}, |\sigma_j(\overline{\mathcal{J}}^{(i)})|^{2-p}\}} \leq \frac{p}{\epsilon^{1-p}}$$

$$\Longrightarrow \left\|\frac{\partial \|\overline{\mathcal{J}}^{(i)}\|_{\circledS \mathfrak{p}}^p}{\partial \overline{\mathcal{J}}^{(i)}}\right\|_F^2 \leq \sum_{i=1}^n \frac{p^2}{\epsilon^{2(1-p)}}.$$

So $\frac{\partial \|\overline{\mathcal{J}}\|_{\circledS \mathfrak{p}}^p}{\partial \overline{\mathcal{J}}}$ is bounded.

Let us denote $\widetilde{\mathbf{F}}_V = \frac{1}{\sqrt{V}} \mathbf{F}_V$, $\mathbf{F}_V$ is the discrete Fourier transform matrix of size $V \times V$, $\mathbf{F}_V^{\mathrm{H}}$ denotes its conjugate transpose. For $\mathcal{J} = \overline{\mathcal{J}} \times_3 \widetilde{\mathbf{F}}_V$ and using the chain rule in matrix calculus, one can obtain that

$$\nabla_{\mathcal{J}} \|\mathcal{J}\|_{\circledS \mathfrak{p}}^p = \frac{\partial \|\mathcal{J}\|_{\circledS \mathfrak{p}}^p}{\partial \overline{\mathcal{J}}} \times_3 \widetilde{\mathbf{F}}_V^{\mathrm{H}}$$

is bounded.

And it follows that

$$\boldsymbol{\mathcal{Y}}_{1,k+1} = \boldsymbol{\mathcal{Y}}_{2,k} + \rho_k(\boldsymbol{\mathcal{H}}_{k+1} - \boldsymbol{\mathcal{J}}_{k+1})$$
$$\Longrightarrow \lambda\nabla_{\boldsymbol{\mathcal{J}}}\|\boldsymbol{\mathcal{J}}_{k+1}\|_{\textcircled{\tiny{$\mathrm{SP}$}}}^p = \boldsymbol{\mathcal{Y}}_{2,k+1},$$

thus $\boldsymbol{\mathcal{Y}}_{2,k+1}$ appears to be bounded.

Moreover, by using the updating rule $\boldsymbol{\mathcal{Y}}_1 = \boldsymbol{\mathcal{Y}}_1 + \mu(\boldsymbol{\mathcal{H}} - \boldsymbol{\mathcal{Q}})$, $\boldsymbol{\mathcal{Y}}_2 = \boldsymbol{\mathcal{Y}}_2 + \rho(\boldsymbol{\mathcal{H}} - \boldsymbol{\mathcal{J}})$, we can deduce $(i = 1, 2)$

$$\begin{aligned}
&\mathcal{L}(\boldsymbol{\mathcal{Q}}_{k+1}, \boldsymbol{\mathcal{G}}_{k+1}, \boldsymbol{\mathcal{H}}_{k+1}, \boldsymbol{\mathcal{J}}_{k+1}; \boldsymbol{\mathcal{Y}}_{i,k}) \\
&\leq \mathcal{L}(\boldsymbol{\mathcal{Q}}_k, \boldsymbol{\mathcal{G}}_k, \boldsymbol{\mathcal{H}}_k, \boldsymbol{\mathcal{J}}_k; \boldsymbol{\mathcal{Y}}_{i,k}) \\
&= \mathcal{L}(\boldsymbol{\mathcal{Q}}_k, \boldsymbol{\mathcal{G}}_k, \boldsymbol{\mathcal{H}}_k, \boldsymbol{\mathcal{J}}_k; \boldsymbol{\mathcal{Y}}_{i,k-1}) \\
&+ \frac{\rho_k + \rho_{k-1}}{2\rho_{k-1}^2}\|\boldsymbol{\mathcal{Y}}_{2,k} - \boldsymbol{\mathcal{Y}}_{2,k-1}\|_F^2 + \frac{\|\boldsymbol{\mathcal{Y}}_{2,k}\|_F^2}{2\rho_k} - \frac{\|\boldsymbol{\mathcal{Y}}_{2,k-1}\|_F^2}{2\rho_{k-1}} \\
&+ \frac{\mu_k + \mu_{k-1}}{2\mu_{k-1}^2}\|\boldsymbol{\mathcal{Y}}_{1,k} - \boldsymbol{\mathcal{Y}}_{1,k-1}\|_F^2 + \frac{\|\boldsymbol{\mathcal{Y}}_{1,k}\|_F^2}{2\mu_k} - \frac{\|\boldsymbol{\mathcal{Y}}_{1,k-1}\|_F^2}{2\mu_{k-1}}.
\end{aligned} \tag{28}$$

Thus, summing two sides of (28) from $k = 1$ to $n$, we have

$$\begin{aligned}
&\mathcal{L}(\boldsymbol{\mathcal{Q}}_{n+1}, \boldsymbol{\mathcal{G}}_{n+1}, \boldsymbol{\mathcal{H}}_{n+1}, \boldsymbol{\mathcal{J}}_{n+1}; \boldsymbol{\mathcal{Y}}_{i,n}) \\
&\leq \mathcal{L}(\boldsymbol{\mathcal{Q}}_1, \boldsymbol{\mathcal{G}}_1, \boldsymbol{\mathcal{H}}_1, \boldsymbol{\mathcal{J}}_1; \boldsymbol{\mathcal{Y}}_{i,0})) \\
&+ \frac{\|\boldsymbol{\mathcal{Y}}_{2,n}\|_F^2}{2\rho_n} - \frac{\|\boldsymbol{\mathcal{Y}}_{2,0}\|_F^2}{2\rho_0} + \sum_{k=1}^n \left(\frac{\rho_k + \rho_{k-1}}{2\rho_{k-1}^2}\|\boldsymbol{\mathcal{Y}}_{2,k} - \boldsymbol{\mathcal{Y}}_{2,k-1}\|_F^2\right) \\
&+ \frac{\|\boldsymbol{\mathcal{Y}}_{1,n}\|_F^2}{2\mu_n} - \frac{\|\boldsymbol{\mathcal{Y}}_{1,0}\|_F^2}{2\mu_0} + \sum_{k=1}^n \left(\frac{\mu_k + \mu_{k-1}}{2\mu_{k-1}^2}\|\boldsymbol{\mathcal{Y}}_{1,k} - \boldsymbol{\mathcal{Y}}_{1,k-1}\|_F^2\right).
\end{aligned} \tag{29}$$

Observe that

$$\sum_{k=1}^\infty \frac{\rho_k + \rho_{k-1}}{2\rho_{k-1}^2} < \infty, \sum_{k=1}^\infty \frac{\mu_k + \mu_{k-1}}{2\mu_{k-1}^2} < \infty,$$

we have the right-hand side of (29) is finite and thus $\mathcal{L}(\boldsymbol{\mathcal{Q}}_{n+1}, \boldsymbol{\mathcal{G}}_{n+1}, \boldsymbol{\mathcal{H}}_{n+1}, \boldsymbol{\mathcal{J}}_{n+1}; \boldsymbol{\mathcal{Y}}_{i,n})$ is bounded. Notice from (7) in the main body

$$\begin{aligned}
&\mathcal{L}(\boldsymbol{\mathcal{Q}}_{n+1}, \boldsymbol{\mathcal{G}}_{n+1}, \boldsymbol{\mathcal{H}}_{n+1}, \boldsymbol{\mathcal{J}}_{n+1}; \boldsymbol{\mathcal{Y}}_{i,n}) \\
&= \sum_{v=1}^V \left\|\bar{\boldsymbol{\mathcal{S}}}^{(v)} - \bar{\boldsymbol{\mathcal{H}}}_{n+1}^{(v)}(\bar{\boldsymbol{\mathcal{G}}}_{n+1}^{(v)})^T\right\|_F^2 \\
&+ \lambda\|\boldsymbol{\mathcal{J}}_{n+1}\|_{\textcircled{\tiny{$\mathrm{SP}$}}}^p + \frac{\rho_n}{2}\|\boldsymbol{\mathcal{H}}_{n+1} - \boldsymbol{\mathcal{J}}_{n+1} + \frac{\boldsymbol{\mathcal{Y}}_{2,n}}{\rho_n}\|_F^2 \\
&+ \frac{\mu_n}{2}\sum_{v=1}^V \|\bar{\boldsymbol{\mathcal{H}}}_{n+1}^{(v)} - \bar{\boldsymbol{\mathcal{Q}}}_{n+1}^{(v)} + \frac{\bar{\boldsymbol{\mathcal{Y}}}_{1,n+1}^{(v)}}{\mu_n}\|_F^2,
\end{aligned} \tag{30}$$

and each term of (30) is nonnegative, following from the boundedness of $\mathcal{L}(\boldsymbol{\mathcal{Q}}_{n+1}, \boldsymbol{\mathcal{G}}_{n+1}, \boldsymbol{\mathcal{H}}_{n+1}, \boldsymbol{\mathcal{J}}_{n+1}; \boldsymbol{\mathcal{Y}}_{i,n})$, we can deduce each term of (30) is bounded. And $\|\boldsymbol{\mathcal{J}}_{n+1}\|_{\textcircled{\tiny{$\mathrm{SP}$}}}^p$ being bounded implies that all singular values of $\boldsymbol{\mathcal{J}}_{n+1}$ are bounded and hence $\|\boldsymbol{\mathcal{J}}_{n+1}\|_F^2$ (the sum of squares of singular values) is bounded. Therefore, the sequence $\{\boldsymbol{\mathcal{J}}_k\}$ is bounded.

Because

$$\boldsymbol{\mathcal{Y}}_{1,k+1} = \boldsymbol{\mathcal{Y}}_{1,k} + \mu_k(\boldsymbol{\mathcal{Q}}_k - \boldsymbol{\mathcal{H}}_k) \Longrightarrow \boldsymbol{\mathcal{H}}_k = \boldsymbol{\mathcal{Q}}_k + \frac{\boldsymbol{\mathcal{Y}}_{1,k+1} - \boldsymbol{\mathcal{Y}}_{1,k}}{\mu_k},$$

and in light of the boundedness of $\boldsymbol{\mathcal{Q}}_k, \boldsymbol{\mathcal{Y}}_{1,k}$, it is clear that $\boldsymbol{\mathcal{H}}_k$ is also bounded.

And from (8) in the main body, it is evident that $\|\bar{\boldsymbol{\mathcal{G}}}_k^{(v)}\|_F^2 \leq \|(\bar{\boldsymbol{\mathcal{S}}}^{(v)})^{\mathrm{T}}\|_F^2 \|\bar{\boldsymbol{\mathcal{H}}}_k^{(v)}\|_F^2$, so $\bar{\boldsymbol{\mathcal{G}}}_k^{(v)}$ is also bounded. So $\boldsymbol{\mathcal{G}}_k$ is bounded.

## A.2 Proof of the 2nd part

From Weierstrass-Bolzano theorem, there exists at least one accumulation point of the sequence $\mathcal{P}_k$. We denote one of the points $\mathcal{P}^* = \{\mathcal{H}^*, \mathcal{Q}^*, \mathcal{G}^*, \mathcal{J}^*, \mathcal{Y}_1^*, \mathcal{Y}_2^*\}$. Without loss of generality, we assume $\{\mathcal{P}_k\}_{k=1}^{+\infty}$ converge to $P^*$.

Note that from the updating rule for $\mathcal{Y}_1$, we have

$$\mathcal{Y}_{1,k+1} = \mathcal{Y}_{1,k} + \mu_k(\mathcal{H}_k - \mathcal{Q}_k) \implies \mathcal{Q}^* = \mathcal{H}^*.$$

Note that from the updating rule for $\mathcal{Y}_2$, we have

$$\mathcal{Y}_{2,k+1} = \mathcal{Y}_{2,k} + \rho_k(\mathcal{H}_k - \mathcal{J}_k) \implies \mathcal{J}^* = \mathcal{H}^*.$$

In the $\bar{\mathcal{G}}^{(v)}$-subproblem (8) in the main body, we have

$$\bar{\mathcal{G}}_k^{(v)} = (\bar{\mathcal{S}}^{(v)})^{\mathrm{T}}\bar{\mathcal{H}}_k^{(v)} \implies \bar{\mathcal{G}}^{(v)*} = (\bar{\mathcal{S}}^{(v)})^{\mathrm{T}}\bar{\mathcal{H}}^{(v)*}.$$

In the $\mathcal{J}$-subproblem (24) in the main body, we have

$$\lambda\nabla_{\mathcal{J}}\|\mathcal{J}_{k+1}\|_{\mathfrak{S}_p}^p = \mathcal{Y}_{2,k} \implies \mathcal{Y}_1^* = \lambda\nabla_{\mathcal{J}}\|\mathcal{J}^*\|_{\mathfrak{S}_p}^p.$$

Therefore, one can see that the sequences $\mathcal{H}^*, \mathcal{Q}^*, \mathcal{G}^*, \mathcal{J}^*, \mathcal{Y}_1^*, \mathcal{Y}_2^*$ satisfy the KKT conditions of the Lagrange function (7) in the main body.

## B  Anchor Selection And Graph Construction

Inspired by [21], we adopt directly alternate sampling (DAS) to select anchors.

First of all, with the given data matrices $\{\mathbf{X}^{(v)}\}_{v=1}^V$, we concatenate the data matrix of each view along the feature dimension. The connected feature matrix $\mathbf{X} \in \mathbb{R}^{n \times d}$ can be represented as $\mathbf{X} = [\mathbf{X}^{(1)}; \mathbf{X}^{(2)}; \cdots; \mathbf{X}^{(v)}]$, where $d$ is the sum of the number of features in each view. Let $\theta_i$ represent the $i$-th sample of the $d$-dimensional features, which can be calculated as

$$\theta_i = \sum_{j=1}^{dT} Tra(X_{ij}), \tag{31}$$

where $dT = \sum_{v=1}^V d_v$, and $Tra(\cdot)$ represents the transformation of the raw features. Specifically, if the features are negative, we process the features of each dimension by subtracting the minimum value in each dimension. Then we obtain the score vector $\boldsymbol{\theta} = [\theta_1, \theta_2, \cdots, \theta_n] \in \mathbb{R}^n$. We choose the point where the maximum score is located as the anchor. The position of the largest score is

$$Index = \arg\max_i \theta_i. \tag{32}$$

Then the 1st anchor of the $v$-th view is $b_1^{(v)} = x_{Index}^{(v)}$.

After that, let $\theta_{Index}$ be the score of the anchor selected from the last round, then we normalize the score of each sample by:

$$\theta_i \leftarrow \frac{\theta_i}{\max\boldsymbol{\theta}}, (i = 1, 2, \cdots, n) \tag{33}$$

Then the score $\theta_i$ can be updated as

$$\theta_i \leftarrow \theta_i \times (1 - \theta_i). \tag{34}$$

Finally, we repeat (32) - (34) $m$ times to select $m$ anchors. After selecting $m$ anchors, we construct an anchor graph of each view $\mathbf{S}^{(v)}$, in the same way, as [21].

# C  More Details of the Experiments

## C.1  Experimental Configurations

The $Reuters$ and $NoisyMNIST$ are implemented on a standard Windows 10 Server with two Intel (R) Xeon (R) Gold 6230 CPUs  2.1 GHz and 128 GB RAM, MATLAB R2020a. The $MSRC$, $HandWritten4$, $Mnist4$ and $AWA$ are implemented on a laptop computer with an Inter Core i5-8300H CPU and 16 GB RAM, using Matlab R2018b.  Codes are available: https://github.com/xdjingli/Orth-NTF.

We repeated the all methods 20 times independently and showed the averages with the corresponding standard deviations. The specific hype-parameters on each dataset are as follows:

- MSRC: anchor rate = 0.7, $p = 0.5$, $\lambda = 100$.
- HandWritten4: anchor rate = 1.0, $p = 0.1$, $\lambda = 1180$.
- Mnist4: anchor rate = 0.6, $p = 0.1$, $\lambda = 5000$.
- AWA: anchor rate = 1.0, $p = 0.5$, $\lambda = 1000$.
- Reuters: anchor rate = 0.005 (anchor number = 100), $p = 0.4$, $\lambda = 1209800$.
- NoisyMnist: anchor rate = 0.03, $p = 0.1$, $\lambda = 200000$.

## C.2  Impact for Parameters

In our proposed algorithm, the number of anchors, the value of $p$ from the tensor Schatten $p$-norm, and the value of $\lambda$ are variable parameters. In this section we take 4 datasets: MSRC, HandWritten4, Mnist4, and AWA as examples to analyze the effect of these variable parameters.

**Effect of the number of anchors.** We changed the anchor rate from 0.1 to 1.0 with step size 0.1. The changes of clustering results and algorithm running time along with the anchor rate were tested on MSRC, HandWritten4, Mnist4 and AWA, as shown in Fig 4 and Fig 5. When the anchor rate were 0.7, 1.0, 0.6 and 1.0, the best clustering results were obtained on MSRC, HandWritten4, Mnist4 and AWA, respectively. The time required for clustering is approximately linearly related to the increase of anchor rate.

**Effect of the value of $p$.** We set the value of $p$ to be 0.1 to 1.0 with a step of 0.1. We obtained the results of ACC, NMI, and Purity in experiments with different values of p as shown in Fig 6. The best clustering results are obtained on MSRC, HandWritten4, Mnist4 and AWA when the values of $p$ are $0.5$, $0.2$, $0.1$, and $0.5$, respectively. It indicates that tensor Schatten pnorm can take advantage of the low-rank of views which helps mine the complementary information of different views. This helps get better clustering results.

**Effect of the value of $\lambda$.** To determine the value of $\lambda$, we initially approximate its range using the magnitude of the tensor Schatten $p$-norm regularization, followed by a more detailed fine-tuning within that range. The impact of varying parameter combinations on the method's performance can be seen in Fig 7. This figure highlights the clustering performance across different pairings of $p$ and $\lambda$.

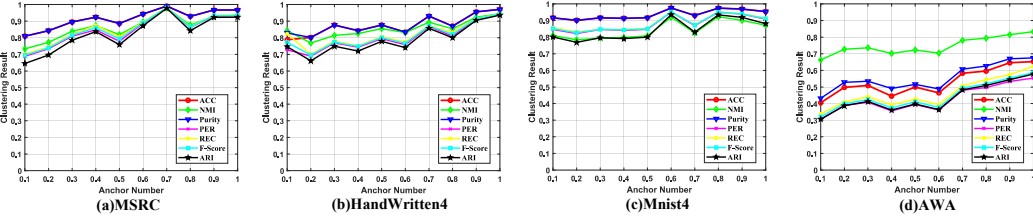

Figure 4: Clustering results with different anchor rate on MSRC, HandWritten4, Mnist4 and AWA.

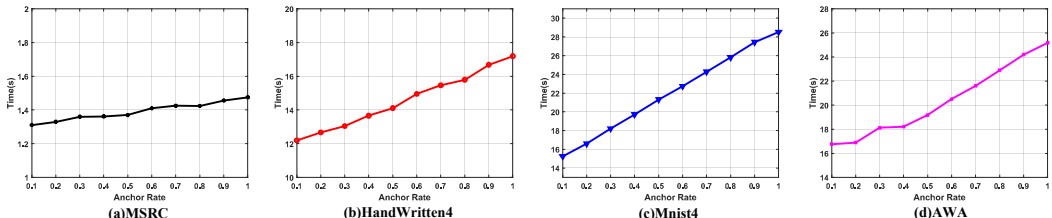

Figure 5: Time (sec.) with different number of anchors on MSRC, HandWritten4, Mnist4 and AWA.

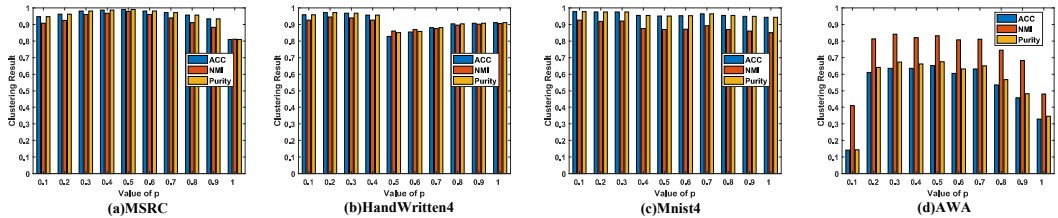

Figure 6: The influence of $p$ on clustering results on MSRC, HandWritten4, Mnist4 and AWA.

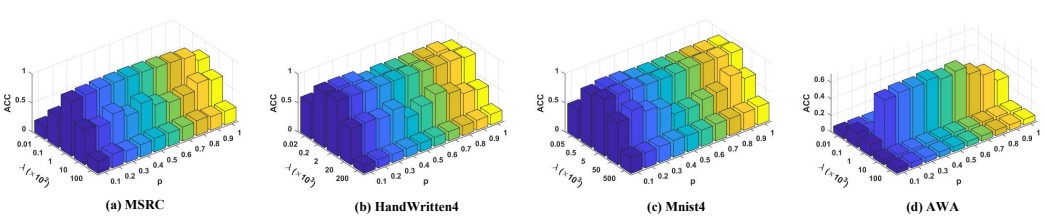

Figure 7: The influence of $\lambda$ and $p$ on clustering results on MSRC, HandWritten4, Mnist4 and AWA.

