# OpenReview forum: "Orthogonal Non-negative Tensor Factorization based Multi-view Clustering"
_NeurIPS.cc/2023/Conference — NeurIPS 2023 poster_

### Official Review · Reviewer_xoPB · 2023-06-28

**Soundness:** 2 fair
**Presentation:** 2 fair
**Contribution:** 3 good
**Rating:** 5
**Confidence:** 4

**Summary:**

Existing NMF-based multi-view clustering methods perform NMF on each view respectively and ignore the impact of between-view. To solve these problems, this paper proposes an orthogonal non-negative tensor factorization method with one-side orthogonal constraint. This method can process the multi-view data directly and can also take full advantage of the original spatial structure of the multi-view data. Extensive experiments on various benchmark datasets indicate that the proposed method can obtain satisfactory clustering performance.

**Strengths:**

(1)	The proposed method can directly consider the between-view relationship and perform Orth-NTF on the 3rd-order tensor which is composed of anchor graphs of views.
(2)	The construction of anchor graph reduces the complexity of the proposed algorithm, while the tensor Schatten p-norm regularization explores the cluster structure of multi-view data.


**Weaknesses:**

(1)	The Introduction and Related work mainly introduce the NMF and its advantages and disadvantages. However, the authors did not adequately explain the motivation for designing the tensor factorization and orthogonal constraint.
(2)	In section 5.2, the author only briefly introduces that their method has achieved good results without any in-depth analysis, which cannot provide any valuable insights for the readers.
(3)	The authors should add more ablation experiments to clearly point out which basic design (or technical idea) of the whole method makes the largest contribution.
(4)	It is not clear how to determine the optimal parameter \lambda in Eq. (6). The authors also failed to analyze in detail the impact of different parameter combinations on the performance of the method, which leads to unconvincing results.


**Questions:**

(1)	The authors mainly introduce the NMF and its variants in Introduction and Related work. However, existing published work on multi-view clustering based on anchor graphs has not been discussed in detail.
(2)	The authors should design more reasonable and complete experiments to prove the effectiveness of the proposed method. In addition, the quality of figures and tables should be improved.


**Limitations:**

The authors need to provide more abundant experimental results to further prove the effectiveness of the proposed method.

---

> ### Author Rebuttal · Authors · 2023-08-08
>
> __Q1__: The Introduction and Related work mainly introduce the NMF and its advantages and disadvantages. However, the authors did not adequately explain the motivation for designing the tensor factorization and orthogonal constraint.
>
> __A1__: Thank you for highlighting that. Non-negative Matrix Factorization (NMF) is tailored primarily for second-order matrices. When processing third-order tensors, there's a need to first transform the tensor into a matrix before applying NMF. This step can lead to a loss of inherent spatial structural information from the third-order tensor. In contrast, Non-negative Tensor Factorization (NTF) sidesteps this issue. NTF directly decomposes third-order tensors, effectively capturing the spatial information they contain.
>
> In the realm of multi-view clustering, conventional NMF-based methods apply NMF to each view independently. Subsequently, they combine the low-dimensional representations from different perspectives to arrive at a unified shared representation. This approach often overlooks the interrelationships between the views, which are crucial for clustering. Our model, however, directly implements NTF on the third-order tensor made up of anchor graphs from the various views. This ensures that the NTF not only acknowledges the relationships between the views but also harnesses the complementary information they offer. __Figure 1 in the provided PDF__ delineates the distinction between traditional NMF-based clustering techniques and our NTF-based approach. Furthermore, by incorporating an orthogonal constraint, our model offers distinct physical interpretability for clustering. This suggests that each row of the indicator matrix contains a single non-zero element, and the position of this element directly corresponds to the label of the respective sample.
>
> __Q2__: In section 5.2, the author only briefly introduces that their method has achieved good results without any in-depth analysis.
>
> __A2__: Thank you for pointing that out. Based on the experimental results presented in our paper, our method significantly outperforms other clustering approaches. This advantage may stem from the fact that our model directly factorizes the tensorized anchor graph—comprised of anchor graphs from various views—into the product of two non-negative tensors, one being an index tensor. As a result, our model effectively captures both the spatial structural information and the complementary data present in the anchor graphs from different perspectives. Additionally, with orthogonal and non-negative constraints in place, our model offers clear interpretability for clustering. This means that each row of the indicator matrix for every view contains a single non-zero element, with its position indicating the label of the associated sample. Consequently, our model can immediately provide the label without necessitating any post-processing, a step which other methods still require.
>
> __Q3__: The authors should add more ablation experiments to clearly point out which basic design (or technical idea) of the whole method makes the largest contribution.
>
> __A3__: Thanks very much for this constructive advice. In the experiments, we added some ablation experiments on orthogonal constraint and Schatten p-norm on four datasets. __Please see Table 1 in provided PDF__.
>
> It can be found that, tensor Schatten p-norm regularization is overall superior to orthogonal constraint. The reason is that tensor Schatten p-norm regularization effectively characterizes both the complementary information and spatial structure information of index matrices of different views. Compared to orthogonal and tensor Schatten p-norm constraints, Joint constraints have great contribution for clustering.
>
> __Q4__: It is not clear how to determine the optimal parameter \lambda in Eq. (6). The authors also failed to analyze in detail the impact of different parameter combinations on the performance of the method.
>
> __A4__: Thank you for bringing that to our attention. To determine the value of $\lambda$, we initially approximate its range using the magnitude of the tensor Schatten p-norm regularization, followed by a more detailed fine-tuning within that range. The impact of varying parameter combinations on the method's performance can be seen __in Figure 3 of the provided PDF__. This figure highlights the clustering performance across different pairings of $p$ and $\lambda$.
>
> __Q5__: The authors introduce the NMF and its variants in Introduction and Related work. However, existing published work on multi-view clustering based on anchor graphs has not been discussed in detail.
>
> __A5__: Thanks very much. Our main contribution of the paper is to propose a non-negative tensor factorization (NTF) model with orthogonal constraint, and then apply it to tensorized graph for large-scale multi-view clustering. Considering the length limitation of paper, we did not discuss the anchor-based clustering methods in detail. It is true that, there have many anchor-based multi-view clustering methods, but all of them separately process the anchor graph of each view respectively, and then fuse them to obtain a common-shared anchor graph. In contrast, our model applies NTF directly to the third-order tensor formed by anchor graphs from different views. As a result, it not only leverages the relationships between the views but also taps into the complementary information they offer. According to your constructive advice, we will add a detail description for anchor-based clustering methods in the revised paper.
>
> __Q6__: The authors should design more experiments. In addition, the quality of figures and tables should be improved.
>
> __A6__: Thank you for your advice. We have added ablation experiments as well as experiments on the impact of different parameter combinations on clustering performance (__See Table 1 and Figure 3 in the provided PDF__). We have adjusted the figures and tables to improve their readability and presentation.

---

> > ### Comment · Reviewer_xoPB · 2023-08-13
> > **Replying to Rebuttal by Authors**
> >
> > Thank you for your detailed rebuttal. Your rebuttal has indeed helped me better understand the proposed approach. Overall, I think that the proposed approach has its distinctions but still not so solid from my viewpoint. The advantage of tesor matrix factorization for multi-view data mining is quite mature, along with the orthogonal property are also well discussed in previous tensor based solution. With a justification of the intuition given now, I'm raising my rating from "Borderline Reject" to "Borderline Accept".

---

> > > ### Author Response · Authors · 2023-08-14
> > >
> > > We appreciate the reviewer for the positive feedback and recognition of our work.

---

> > > > ### Author Response · Authors · 2023-08-18
> > > >
> > > > It is true that Tensor factorization is quite mature in the field of multi view data mining, but these studies mainly focus on CP factorization, Tucker factorization, and high-order singular value decomposition. These factorization involve module k-tensor and matrix multiplication, which destroys the intrinsic structure of tensors. Unlike these tensor factorization, the tensor factorization  our paper studies belongs to Tubal factorization and directly focuses on tensor by tensor multiplication, which is currently a hot research topic, especially for research on both non-negative and orthogonal constraints, there are few reports. This papers focus on this problem and develops a new model for  multi-view clustering. Thanks your positive feedback and recognition of our work.

---

### Official Review · Reviewer_J3E2 · 2023-07-03

**Soundness:** 3 good
**Presentation:** 3 good
**Contribution:** 3 good
**Rating:** 7
**Confidence:** 5

**Summary:**

This paper presented an orthogonal semi-nonnegative tensor factorization and proposed a novel tensorized anchor graph factorization model for Multiview clustering. Compared with existing NMF-based multi-view clustering methods, the proposed model has the following advantages: First, the proposed model directly factorizes 3-order tensorized anchor graph for clustering, while existing methods perform NMF on each view respectively. Thus the proposed method well exploits the within-view and between-views spatial structure information. Second, the proposed model has good interpretability for clustering with orthogonal non-negative constraint on tensorized soft labels of views. Third, authors use the tensor Schatten p-norm regularization as a rank approximation of the 3rd-order tensor which characterizes the cluster structure of multi-view data and exploits the between-view complementary information. Fourth, the paper presented a convergence analysis in theory. Experimental results indicate the efficiency of the proposed model on some databases.

**Strengths:**

(1)	The paper presented an orthogonal semi-nonnegative tensor factorization.

(2)	The paper proposed a novel tensorized anchor graph factorization model for Multiview clustering with good interpretability for clustering by orthogonal non-negative constraint on tensorized soft labels of views, this avoids post-processing for clustering.

(3)	The proposed model directly factorized 3-order tensorized anchor graph for clustering, while existing methods perform NMF does not.

(4)	The paper mathematically proved the convergence of the proposed algorithm for clustering.


**Weaknesses:**

(1) The paper does not provide the storage complexity and computational complexity.

(2) It is unclear how to select anchor points or construct anchor graph.

(3) It is unclear for the variables in (19).

(4) In algorithm 1, mu and \pho will be infinity. This will make the algorithm be unstable.

(5) In the experiments, how to select \ eta and \ lambda?

(6) Figure 6 indicates that the proposed model has good convergence, how does the performance (such as ACC) of the proposed model change?


**Questions:**

(1) In the paper, how to select anchor points or construct anchor graph?

(2) In algorithm 1, mu and \pho will be infinity. This will make the algorithm be unstable.

  (3) What is the storage complexity and computational complexity for the model?

 (4) It is unclear for the variables in (19).


**Limitations:**

   Authors adequately addressed the limitations of the existing NMF-based multi-view clustering methods.

---

> ### Author Rebuttal · Authors · 2023-08-08
>
> __Q1__: The paper does not provide the storage complexity and computational complexity.
>
> __A1__: Thank you for the clarification. For Orth-NTF, the storage requirements for $\mathcal{G}$, $\mathcal{Q}$, $\mathcal{H}$, $\mathcal{J}$, $\mathcal{Y}_1$, and $\mathcal{Y}_2$ have complexities of $\mathcal{O}(V(m+k)n)$, $\mathcal{O}(V(n+k)k)$, $\mathcal{O}(Vnk)$, $\mathcal{O}(Vnk)$, $\mathcal{O}(Vnk)$, and $\mathcal{O}(V(n+k)k)$, respectively. Combining these, the total storage complexity for Orth-NTF is $\mathcal{O}(Vnm+6Vnk+2vk^2)$.
>
> The process of constructing $\mathcal{S}$ has a computational complexity of $\mathcal{O}(Vnmd+Vnm\log(m))$. When updating the four variables—G, Q, H, and J—their respective computational complexities are $\mathcal{O}(Vnkm+Vnk\log(k))$, $\mathcal{O}(Vm^2 k+Vmk^2)$, $\mathcal{O}(Vnk)$, and $\mathcal{O}(2Vnk\log(Vk)+V^2 kn)$. Given that $m$, $n$, $k$, and $V$ are relatively small constants, the primary computational cost associated with updating the variables stands at $\mathcal{O}(Vnkm+Vm^2 k)$. Summing it all up, the overall computational complexity of our proposed method is $\mathcal{O}(Vnmd+Vm^2 k)$.
>
> __Q2__: It is unclear how to select anchor points or construct anchor graph.
>
> __A2__: Thanks very much. We adopt directly alternate sampling (DAS) to select anchors inspired by [1], and we construct anchor graph in the same way as [1]. We also explain about anchor selection and anchor graph construction in our Supplementary Material.
>
> [1] Li, X., Zhang, H., Wang, R., and Nie, F. Multiview clustering: A scalable and parameter-free bipartite graph fusion method. IEEE Transactions on Pattern Analysis and Machine Intelligence, 44(1):330–344, 2022.
>
> __Q3__: It is unclear for the variables in (19).
>
> __A3__: Thanks very much. Sorry for the confusion. $\overline{\Lambda}^{(v)}$ and $\overline{V}^{(v)}$ can be obtained by SVD($\overline{\mathcal{B}}^{(v)}$) = $\overline{\Lambda}^{(v)} X {\overline{V}^{(v)}}^T$, where $\overline{\mathcal{B}}^{(v)}$=$ 2\overline{\mathcal{S}}^{(v)}$ $\overline{\mathcal{G}}^{(v)}$+$\mu \overline{\mathcal{H}}^{(v)}$ + $\overline{\mathcal{Y}}_1^{(v)}$.
>
> __Q4__: In algorithm 1, mu and \pho will be infinity. This will make the algorithm be unstable.
>
> __A4__: Thanks very much. In the experiments, we set the maximum of mu and $\rho$ to $10^{13}$. We have explicitly pointed out this in our revised paper.
>
> __Q5__: In the experiments, how to select \ eta and \ lambda?
>
> __A5__: Thank you for noting that. The value of $\eta$ influences the algorithm's convergence speed, and based on our empirical observations, we set $\eta$ = 1.6. As for $\lambda$, we initially estimate its range by considering the magnitude of the tensor Schatten $p$-norm regularization and subsequently fine-tune within that established range.
>
> __Q6__: Figure 6 indicates that the proposed model has good convergence, how does the performance (such as ACC) of the proposed model change?
>
> __A6__: Thanks very much.
> When the number of iteration increases, the clustering metric (such as ACC) overall improves gradually and tends to be constant with the convergence of the algorithm (__See Figure 2 in provided PDF__). It also indicates that our method has good clustering performances.

---

> > ### Comment · Reviewer_J3E2 · 2023-08-13
> > **Review Results**
> >
> > I have carefully read the rebuttal and I think all my concerns have been well addressed. So I am willing to keep the final rating as ACCEPT.

---

> > > ### Author Response · Authors · 2023-08-14
> > >
> > > We appreciate the reviewer for the positive feedback and recognition of our work.

---

### Official Review · Reviewer_PAbp · 2023-07-04

**Soundness:** 3 good
**Presentation:** 3 good
**Contribution:** 3 good
**Rating:** 7
**Confidence:** 5

**Summary:**

This article proposed a novel orthogonal non-negative tensor factorization strategy for multi-view clustering, which well takes into account within-view spatial structure and between-view complementary information. Meanwhile, the optimization step has good convergency.

**Strengths:**

[a] The paper is well-written and easy to follow.
[b] The proposed model is concise and has good interpretability.
[c] The experimental results are substantial.

**Weaknesses:**

1. Some formulas are not strictly written, which variable to solve should be clearly written.
2. What does each letter of the matrix size represent?
3. In Algorithm 1, how to calculate the clustering label is not clear.
4. Authors are advised to report specific hyper-parameters on each dataset.
5. In Section 5.3, the author draws a conclusion that the clustering time increases linearly with the increase of anchor rate. This description seems imprecise.
6. Some typos, for example:
  In Section 5.3, the Schatten p norm should be ‘Schatten p norm.’
  In Section 5.2, ‘according to Tables 2 and 3’ should be ‘according to Tables 2 and 3.’

**Questions:**

1.Some formulas are not strictly written, which variable to solve should be clearly written.
2.What does each letter of the matrix size represent?
3. In Algorithm 1, how to calculate clustering label is not clear.
4. Authors are advised to report specific hyper-parameters on each dataset.
5. In Section 5.3, the author draws a conclusion that the clustering time increases linearly with the increase of anchor rate. This description seems imprecise.

**Limitations:**

1. The source code and datasets are encouraged to release.
2. It is encouraged that the author can discuss the practical application scenarios of multi-view clustering technology.

---

> ### Author Rebuttal · Authors · 2023-08-08
>
> __Q1__: Some formulas are not strictly written, which variable to solve should be clearly written.
>
> __A1__: Thanks very much. We double checked our manuscript and corrected formulas to explicitly indicator which variables need to be solved.
>
> __Q2__: What does each letter of the matrix size represent?
>
> __A2__: Thanks very much. In our article, $n$, $m$ and $k$ represent the number of samples, the number of anchors and the number of clusters, respectively. We have explicitly pointed out the meaning of each letter of the matrix size in the revised paper.
>
> __Q3__: In Algorithm 1, how to calculate the clustering label is not clear.
>
> __A3__: Thanks very much. The position of the largest element in each row of the indicator matrix is the label of the corresponding sample. We have explicitly pointed out this in our revised paper.
>
> __Q4__: Authors are advised to report specific hyper-parameters on each dataset.
>
> __A4__: Thanks very much. The corresponding hyper-parameters for each dataset are as follows: __MSRC__: anchor rate=0.7, $p$=0.5, $\lambda$=100; __HandWritten4__: anchor rate=1.0, $p$=0.1, $\lambda$=1180; __Mnist4__: anchor rate=0.6, $p$=0.1, $\lambda$=5000; __AWA__: anchor rate=1.0, $p$=0.5, $\lambda$=1000; __Reuters__: anchor rate=0.005(anchor number=100), $p$=0.4, $\lambda$=1209800; __NoisyMNIST__: anchor rate=0.03, $p$=0.1, $\lambda$=200000. We have explicitly pointed out this in our revised paper.
>
> __Q5__: In Section 5.3, the author draws a conclusion that the clustering time increases linearly with the increase of anchor rate. This description seems imprecise.
>
> __A5__: Thanks very much. I corrected these inaccurate representations. It should be that the time required for clustering is approximately linearly related to the increase of anchor rate.
>
> __Q6__: Some typos, for example: In Section 5.3, the Schatten p norm should be ‘Schatten p norm.’ In Section 5.2, ‘according to Tables 2 and 3’ should be ‘according to Tables 2 and 3.’
>
> __A6__: Thanks very much. We double checked the manuscript and corrected them.
>
> __Q7__: The source code and datasets are encouraged to release.
>
> __A7__: Thanks. We're sorry we can't use any links in our reply. __We've sent an anonymized link to the AC as required__. The datasets we used is open source.
>
> __Q8__: It is encouraged that the author can discuss the practical application scenarios of multi-view clustering technology.
>
> __A8__: Thank you for pointing that out. Multi-view clustering techniques have been applied in a myriad of practical situations across diverse fields. To highlight a few applications:
>
> 1. In social media analysis, multi-view clustering allows for grouping based on textual content, visual features, and user network structures, thereby aiding in community identification or anomaly detection.
> 2. Recommender systems leverage this clustering technique across various user and item views, which can enhance the personalization, accuracy, and diversity of their recommendations.

---

### Official Review · Reviewer_a6FX · 2023-07-06

**Soundness:** 3 good
**Presentation:** 4 excellent
**Contribution:** 2 fair
**Rating:** 5
**Confidence:** 4

**Summary:**

In this paper, the authors focus on the problem of multi-view clustering using semi-non-negative tensor factorization (Orth-NTF) with a one-side orthogonal constraint. The proposed model extends Non-negative Matrix Factorization (NMF) to Orth-NTF, allowing for the utilization of spatial structure information from multi-view data to enhance clustering performance. Moreover, the authors fully leverage the complementary information embedded in different views by incorporating a tensor Schatten p-norm composed of cluster indicator matrices. To reduce computational complexity, anchor graphs are adopted instead of the original multi-view data. The authors provide an optimization algorithm for the proposed method and demonstrate its effectiveness through extensive experiments conducted on various datasets.

**Strengths:**

1.The overall structure of the paper is clear and comprehensive.
2.The research problem and innovative aspects are well-defined, and they are supported by sufficient experimental evidence.
3.The proposed method demonstrates promising results for the task of multi-view clustering.
4.The provided optimization method includes detailed formula derivation, enhancing the understanding of the proposed approach.

**Weaknesses:**

1.The paper seems to lack a detailed explanation of the advantages of extending NMF to 3rd-order tensor NMF.
2.The proposed method applies NMF to the anchor graph S. Couldn't other dimensionality reduction methods be used to learn low-dimensional representations of high-dimensional data and achieve acceleration in clustering? Why was the choice made to utilize the anchor graph?
3.I think the author should provide the code to make the experimental results more convincing.

**Questions:**

Please refer to the weakness section.

---

> ### Author Rebuttal · Authors · 2023-08-08
>
> __Q1__: The paper seems to lack a detailed explanation of the advantages of extending NMF to 3rd-order tensor NMF.
>
> __A1__: Thank you for your attention. Non-negative Matrix Factorization (NMF) is designed primarily for second-order matrices. When handling third-order tensors, one must first transform the tensor into a matrix to apply NMF. This transformation leads to a loss of spatial structural information inherent in the third-order tensor. Conversely, Non-negative Tensor Factorization (NTF) is adept at directly decomposing third-order tensors, effectively preserving the spatial information they contain.
>
> For multi-view clustering, traditional NMF-based methods execute NMF separately on each view. They then amalgamate the low-dimensional representations from different perspectives to derive a common shared representation. However, these methods tend to overlook the interrelationships between the views, which play a crucial role in clustering. Our model, in contrast, applies NTF directly to the third-order tensor formed by anchor graphs from different views. As a result, it not only leverages the relationships between the views but also taps into the complementary information they offer.
>
> To visually understand the distinction between prevailing NMF-based clustering techniques and our NTF-based model, __please refer to Figure 1 in the provided PDF__.
>
> __Q2__: The proposed method applies NMF to the anchor graph S. Couldn't other dimensionality reduction methods be used to learn low-dimensional representations of high-dimensional data and achieve acceleration in clustering? Why was the choice made to utilize the anchor graph?
>
> __A2__: Thank you for highlighting that. While other dimensionality reduction methods like PCA and LPP can indeed be utilized to derive low-dimensional representations and handle large-scale data effectively, NMF offers certain unique advantages. Compared to PCA and LPP, NMF boasts superior interpretability. When augmented with an orthogonal constraint, NMF gains even clearer physical interpretability in the realm of clustering. This means that each row of the indicator matrix possesses just one non-zero element, with the position of that element directly signifying the label of the corresponding sample. Additionally, the anchor graph excels in clustering due to its ability to adeptly encapsulate relationships between data points of any shape. Inspired by these strengths, we opted for anchor graph in our approach to large-scale multi-view clustering.
>
> __Q3__: I think the author should provide the code to make the experimental results more convincing.
>
> __A3__: Thanks very much. We're sorry we can't use any links in our reply. __We've sent an anonymized link to the codes to the AC as required__.

---

> > ### Comment · Reviewer_a6FX · 2023-08-18
> >
> > The author's response completely addressed my concerns, and I believe this article should be accepted.

---

> > > ### Author Response · Authors · 2023-08-18
> > >
> > > We appreciate the reviewer for the positive feedback and recognition of our work.

---

### Author Rebuttal · Authors · 2023-08-08

Supplementary PDF is uploaded here as required.

---

### Comment · Area_Chair_LM1t · 2023-08-13
**post-rebuttal review**

Dear Reviewers,

Thanks for your time and efforts in reviewing this paper. The author rebuttal is now available. Please read the rebuttal and update your comments ASAP. Thank you very much!

Best,
AC

---

### Decision · Program_Chairs · 2023-09-21

**Decision:**

Accept (poster)

**Comment:**

After the author-review discussions, all reviewers agree accpeting the paper. I would ecournage the authors to improve their paper according to the reviews.